# Marine mammal tracks from two-hydrophone acoustic recordings made with a glider

Elizabeth T. Küsel[1], Tessa Munoz[1], Martin Siderius[1], David K. Mellinger[2], and Sara Heimlich[2]

[1]Northwest Electromagnetics and Acoustics Research Laboratory, Portland State University, 1900 SW 4th Ave, Portland, OR 97201
[2]Cooperative Institute for Marine Resources Studies, Oregon State University, 2030 Marine Science Drive, Newport, OR 97365

*Correspondence to:* Elizabeth T. Küsel (ekusel@pdx.edu)

**Abstract.** A multinational oceanographic and acoustic sea experiment was carried out in the summer of 2014 off the western coast of the island of Sardinia, Mediterranean Sea. During this experiment, an underwater glider fitted with two hydrophones was evaluated as a potential tool for marine mammal population density estimation studies. An acoustic recording system was also tested, comprising an inexpensive, off-the-shelf digital recorder installed inside the glider. Detection and classification of sounds produced by whales and dolphins, and sometimes tracking and localization, are inherent components of population density estimation from passive acoustics recordings. In this work we discuss the equipment used as well as analysis of the data obtained, including detection and estimation of bearing angles. A human analyst identified the presence of sperm whale (*Physeter macrocephalus*) regular clicks as well as dolphin clicks and whistles. Cross-correlating clicks recorded on both data channels allowed for the estimation of the direction (bearing) of clicks, and realization of animal tracks. Insights from this bearing tracking analysis can aid in population density estimation studies by providing further information (bearings), which can improve estimates.

## 1  Introduction

Autonomous underwater vehicles (AUVs) such as gliders are being used ever more frequently as a tool in ocean research. A glider moves through the water column in a see-saw pattern by controlling its buoyancy, enabling it to glide forward with the use of horizontally-mounted wings. Given their mode of operation, gliders provide a platform that is acoustically very quiet, making them well-suited for passive acoustic monitoring of marine mammals. Increasing amounts of marine mammal recordings are being obtained by fitting gliders with hydrophones (e.g. Klinck et al., 2012; Baumgartner et al., 2013).

Gliders have most often been fitted with a single hydrophone, and recordings from both mysticetes (baleen whales) and odontocetes (toothed whales, dolphins, and porpoises) have been made in this manner. More specifically, beaked whales (Ziphiidae sp.), sperm whales (*Physeter macrocephalus*), and delphinids (Delphinidae sp.), which all produce highly broad-band high-frequency echolocation clicks, were detected in real time from a glider off Hawai'i in 2009 in a study of habitats and vocalization behavior (Klinck et al., 2012). In addition, sei whale (*Balaenoptera borealis*) vocalizations were recorded by a glider to study their diel vocalization patterns (Baumgartner and Fratantoni, 2008). The ability of gliders to perform and report

real-time detections of four different kinds of baleen whales and their different call types has also been tested successfully by Baumgartner et al. (2013).

Estimating population size, or density, of marine mammals from passive acoustic data is a growing research area. Methodologies are still being developed which would apply to data recorded by different passive acoustic sensor arrangements (e.g. fixed vs. towed platforms, single-sensor vs. arrays). The works of Thomas and Marques (2012) and Marques et al. (2013) provide good summaries of the current state of density estimation techniques from passive acoustics applied to different species of marine mammals. Küsel et al. (2011) addressed density estimation from single fixed sensors in which no information on animal location is readily available from the data. In such cases, a modeling approach is used to estimate detection distances, which are then translated into a relationship expressing the probability of detection as a function of range - the *detection function*. More recently, the single-sensor modeling technique was revised and a new approach was suggested for handling cases when the call's bandwidth is on the order of tens of kilohertz (Küsel et al., 2016). While density estimation techniques for data recorded by fixed sensors already exist, the same is not true for slow-moving platforms such as gliders. Research on the extension of density estimation techniques to underwater gliders has therefore become a current research topic given the increasing use of this platform for studying marine mammals. Estimating the detection function is one of the main requirements of density estimation methods. When dealing with data from a slow-moving sensor, the primary issues are the movement of the animals relative to the glider (Glennie et al., 2015), and the slow movement of the glider relative to the speed of whales and dolphins. However, in any kind of density estimation survey, the more information that is available about the animals, the better the density inferences (Borchers et al., 2015).

Therefore, the main objectives of this work were (1) to evaluate the use of an ocean glider for marine mammal population density estimation studies, and (2) to apply the extra information provided by having two sensors separated by a small distance. Two hydrophones can provide bearing angles to vocalizing animals, which in turn can be used as an extra covariate in density estimations. This has been shown to improve inferences for terrestrial wildlife populations (Borchers et al., 2015). Bearing angles can also provide tracks that indicate animal movement. The ability to resolve different animals from estimated tracks can give an idea of the number of animals detected. Finally, tracks can also be used to estimate call production rate. This is species-dependent and is used to estimate how often animals produce calls on average in a day. The call rate is often another important parameter in density estimation (Marques et al., 2009), but accurate call rates remain lacking for most marine mammals.

Some advantages of using a glider for passive acoustic monitoring of marine mammals and density estimation studies include the *in-situ* measurement of conductivity, temperature and depth from which sound speed can be calculated to aid in the estimation of detection distances, and the possibility of mounting multiple hydrophones on the platform, which can yield information on animal location. Gliders also offer an acoustic quiet platform, with short and discrete noisy periods that are easily distinguishable and hence, can be removed from data analysis. In addition, moving sensors such as gliders have an advantage over fixed sensors since they can be relocated as needed and can cover a larger geographic area. Gliders also offer the possibility of reporting data in near-real time, which can help guide field surveys. They are easy to deploy and recover, and

can remain at sea for weeks or even months at a time. One possible disadvantage is the presence of strong currents in a study area, which can move the glider off its planned course.

This work is organized as follows. Section 2 describes the acoustic recording system used in the glider, and the sea experiment. The data processing and analysis, including examples of sounds recorded during the sea trial, are presented in Sec. 3. A brief description of the acoustic environment in the survey area is presented in Sec. 4. Section 5 discusses the tracking results and Sec. 6 discusses the overall work and draws conclusion for future experiments and future work in terms of density estimation applications.

## 2 Methodology

### 2.1 Acoustic Recording System

The Northwest Electromagnetic and Acoustics Research Laboratory (NEAR-Lab) at Portland State University (PSU), Portland, OR, owns a first-generation, 200-meter Webb-Teledyne Slocum glider (Webb et al., 2001) named *Clyde*. Clyde was fitted with two hydrophones (High-Tech HTI-92-WB, with pre-amplifiers). The hydrophones, with sensitivities of -159 and -161.4 $\mathrm{dB}$ re 1 V/µPa, were mounted on the wings of the glider at a horizontal separation of approximately 0.9 $\mathrm{m}$ (Fig. 1(a)).

An inexpensive, off-the-shelf, linear pulse code modulation (PCM) recorder (Tascam DR-07 MKII) was adapted to fit inside the glider's science bay as a stand-alone sensor. It was not connected to the glider's computer, and was independent of glider operations. The recorder was equipped with enough batteries (8 AA alkaline) to record continuously at 96 $\mathrm{kHz}$ sampling frequency and 16-bit resolution for up to 24 hours (Fig. 1(b)). In its original configuration, the Tascam took two AA batteries and recorded sounds by default at 44.1 $\mathrm{kHz}$ and 16-bit resolution. The recorder allowed only continuous recording. The maximum recording time of 24 hours was a function not only of power consumption but also of available storage. Data was recorded to a single micro-SD card, for which the maximum capacity could not exceed 32 GB. A noise assessment of the Tascam was made when it was first acquired and showed higher self-noise at lower frequencies (< 1 kHz). However, the noise was not deemed sufficiently high to consider it a problem. Research for off-the-shelf recorders at the time (2013-2014) indicated that the Tascam offered the highest sampling frequency, while other pocket recorders had sampling frequencies of only up to 44.1 or 48 $\mathrm{kHz}$. Moreover, the acquisition system offered the capability of recording two channels of data, one from each hydrophone.

Testing of the acoustic recording system and data collection took place during an opportunistic sea-trial. No specific marine mammal species were targeted during this experiment. It was understood however, that the system would only be able to detect sounds up to 48 $\mathrm{kHz}$, or half the sampling frequency. While such bandwidth would not be enough to capture all frequencies of, for example, dolphin clicks, it was enough to detect dolphins, potentially classify some of them, and detect and classify other whale species such as sperm whales.

## 2.2  Sea Trial

The sea trial Recognized Environmental Picture 2014 (REP14-MED) took place from 6 to 26 June 2014 in the Western Mediterranean Sea. Its objective was to obtain environment knowledge and uncertainty (geographical, meteorological, oceanographic and acoustic) to support North Atlantic Treaty Organization (NATO) operations. Two vessels participated in the 2014 campaign, the NATO research vessel (NRV) *Alliance* and the German research vessel *Planet*. During the experiment, both physical oceanography and acoustic data were collected, although acoustic experiments were conducted only from the NRV Alliance (Onken et al., 2016).

As part of the experiments, 10 gliders were assigned parallel tracks along an east-west direction perpendicular to the west coast of the island of Sardinia, Italy. Our glider was assigned the northernmost track, and was deployed at $40°$ 00' N $07°$ 22' E at 12:16 Central European Summer Time (CEST) on June 09, 2014 (Fig. 2). It was programmed to dive between 15 and 170 m in the see-saw pattern typical of Slocum gliders at an angle of 26 degrees. It was also initially programmed to surface every 2 hours to send navigation data back to the glider pilots at NATO's Centre for Maritime Research and Experimentation (CMRE) in La Spezia, Italy. In the absence of strong currents, a correctly ballasted Slocum glider can travel at speeds of approximately 0.25 m/s.

Data recording was initiated about one hour prior to deployment while the glider was still on board the NRV Alliance, and ended when the 32 GB micro-SD card inside the recorder was full, approximately 23 hours later. A total of 15 acoustic files containing 22 hours of 2-channel continuous data were recorded between June 09-10 when the glider was located in deep waters (deeper than 2000 m) off of the west coast of Sardinia (Fig. 2).

During a mission far from ship- or land-based radio transponders, gliders communicate at pre-designated surfacing points via Iridium satellite. Communications with Clyde were completely lost around 23:10 CEST on June 10, after acoustic recording had terminated. Its location was re-found only on June 11 around 21:20 CEST via an emergency location beacon in the glider that communicates through a separate (Argos) satellite system. The glider was finally sighted at 17:36 CEST on June 12 at $40°$ 03' N $07°$ 34' E and recovered shortly thereafter, at 17:47, by RV *Planet* (Fig. 2). A hardware malfunction caused not only the loss of communications but also the loss of some navigation files and CTD (conductivity, temperature, and depth) information. Fortunately, the data files for the period when the acoustic recorder was on were intact.

## 3  Data Processing and Analysis

The acoustic data was saved by the Tascam recorder in WAVE (.wav) audio file format. Of the 15 files recorded, 14 had a duration of 1:33:09 hours, while the last file filled the remaining storage and had a duration of 1:20:17 hours. The glider was deployed 1:43:58 hours after the beginning of recordings, implying that the first file contained only recordings made above water. After a glider deployment, a series of test dives are performed to check on the overall functionality and ballasting of the vehicle. Therefore, most of the second file contained recordings made while the glider either made shallow dives or was at the surface. It also appears from the acoustic data that the glider started its primary mission, navigating to its pre-assigned west-east track perpendicular to the coast of Sardinia (Fig. 2), approximately 40 minutes after the actual deployment. The

remainder of the data in the second file did not show any significant marine mammal events. Discounting the first two files, a total of approximately 19.9 hours of data were available for analysis.

The different sounds observed in the acoustic data are presented below. They include marine mammal sounds as well as glider self-noise and other electronic noise, which can potentially impact the data analysis. A description of the glider navigation data and environmental data collected by its conductivity, temperature, and depth (CTD) sensor at the time the acoustic data was recorded is also presented. Such data can help in understanding detection probabilities and the acoustic environment, and are thus important to population density estimation.

### 3.1    Marine Mammal Sounds

According to Notarbartolo di Sciara (2002), 21 species of cetaceans occur in the Mediterranean and Black Seas. Of these, eight species are considered common or regular to the Mediterranean Sea: fin whale (*Balaenoptera physalus*), sperm whale (*Physeter macrocephalus*), Cuvier's beaked whale (*Ziphius cavirostris*), long-finned pilot whale (*Globicephala melas*), Risso's dolphin (*Grampus griseus*), common bottlenose dolphin (*Tursiops truncatus*), striped dolphin (*Stenella coeruleoalba*), and short-beaked common dolphin (*Delphinus delphis*). Minke whale (*B. acutorostrata*), killer whale (*Orcinus orca*), false killer whale (*Pseudorca crassidens*), and rough-toothed dolphin (*Steno bredanensis*) can also occasionally be encountered (Pavan and Borsani, 1997).

Of the cetacean species commonly present in the Mediterranean, striped dolphins are the most abundant (Notarbartolo di Sciara et al., 2008). In terms of the sounds they produce, some species (e.g., fin and sperm whales) have been better studied than others (e.g., long-finned pilot whale). For density estimation purposes, calls that are easily detectable and distinguishable are preferred. One such type of calls is the impulsive and broadband echolocation click, produced by all odontocetes that have been studied acoustically.

Preliminary analysis of the recorded data involved visual inspection of spectrograms by a trained marine bioacoustician to identify marine mammal calls. Results presented in this work were derived from file 06 only, recorded between 19:47 and 21:20 (CEST) on June 09, 2014. The intention here is to demonstrate the type of analyses that could be done with the data recorded from two hydrophones, and not to describe the data set in its entirety. File 06 was chosen due to the extent of marine mammal activity and also due to the fact that the glider did not surface during its recording, providing roughly 1:30 hours of uninterrupted data. This does not imply, however, that there were no data on the remainder of the recordings. In fact, the absence of detected calls is also important in population density estimation. Manual inspection of this file identified sperm whale clicks as well as clicks and whistles from one or more unknown species of dolphins.

Representations of sperm whale echolocation clicks and clicks and whistles from dolphins from the data set are shown in Figs. 3 and 4. Spectrograms continue to be the preferred tool used by many marine bio-acousticians to show snippets of data or detections of marine animal sounds. They give not only the time of occurrence (horizontal axis) but also the frequency content of the call (vertical axis) as well as its energy content (color, usually in decibels). It is noted that if a sound's maximum frequency is larger than half the sampling frequency of the instrument, then it will appear clipped in the spectrogram. With the

sampling frequency of 96 kHz used in this experiment sounds above 48 kHz could not be detected. This can be observed in the spectrogram of Fig. 4, in which dolphin clicks appear truncated at 48 kHz at the top of the plot.

The remainder of the analyses presented here are based on the recordings of sperm whale clicks. Sperm whales produce broadband regular clicks, also called usual clicks (Whitehead and Weilgart, 1990), that are highly directional (Møhl et al., 2000). Their clicks range in frequency from 200 Hz to 32 kHz (Madsen et al., 2002), with center frequency reported around 13.4 kHz (Møhl et al., 2003), inter-click intervals (ICIs) of 0.5 - 2 s (Fig. 3), and duration of 10 - 20 ms (Goold and Jones, 1995). Although no estimate of population size exists for Mediterranean Sea sperm whales, the population is believed to be in decline, with numbers in the hundreds of animals (Notarbartolo di Sciara et al., 2006). So, even though this species has been well studied acoustically, its distribution and occurrence are still not understood as well. With regards to their presence in the study area, the closest account was given by Gannier et al. (2002), who conducted a four-year effort that combined both visual and acoustic methods to study the distribution of sperm whales in a large portion of the Mediterranean Sea. They reported whales concentrating in the surroundings of the Balearic Islands, and to a lesser extent, in the western continental slope off Sardinia (to the north of the study area).

Possible multipath clicks were also observed among sperm whale regular click detections, and an example is shown in Fig. 5 from a 6-second segment of data recorded on channel 1. Multipath occurrence, of any underwater signal, will depend on the geographic location, water column structure, and depth of source. In the case of marine mammal calls, the location and distance of the animals with respect to the recording sensor are not known a priori. Multipath can sometimes be used to aid in localizing whales (e.g. Laplanche et al., 2005). However, in order to automatically distinguish multipath in the recorded data, highly specialized algorithms are necessary. Another option is for a human analyst to manually check the data, which can be a time-consuming task. For density estimation studies, detectors of simple characterization are preferred. Therefore, the use of complex algorithms for selecting only direct arrivals was beyond the scope of this work. Our intent was not to localize animals; being able to resolve tracks is sufficient and less time-consuming for density estimation purposes.

## 3.2   Electronic Noise

Evaluation of data spectrograms and power spectral density plots indicated an increase in energy content at frequencies above 25 kHz (Fig. 6). This increase in power with frequency was considered an artifact, given the well-known increase in attenuation with frequency in the ocean that typically causes a decrease in ambient noise with frequency (Jensen et al., 2011). Moreover, no known physical phenomena would produce the observed elevated noise levels at high frequencies.

Another feature observed in the data set was the presence of high-amplitude, impulse-like spikes, or *glitches*. These features were conspicuously present throughout channel 2, but at lower intensity, and not simultaneously, in channel 1. Even though glitches resembled marine mammal clicks at first glance, whether looking at the time series or spectrograms, closer inspection revealed a characteristic shape and sound suggestive of an electronic artifact produced by the acoustic acquisition system. It was observed in spectrograms that the lower-bound frequency of spikes was 0 Hz, unlike marine mammal clicks, which had a lower bound in the hundreds of hertz or above. Figure 7 shows the same 30 seconds of data recorded on channel 1 and on channel 2. It illustrates the frequency with which glitches occur in each channel, and how they differ in a spectrogram from

sperm whale echolocation clicks. The characteristic signature of glitches in the time domain is illustrated in Fig. 8. It is noted that while glitches can appear with very high amplitudes, sometimes they can also have lower amplitudes comparable to marine mammal sounds. This is also observed in Fig. 8. However, their signature shape is always the same.

## 3.3 Glider Sounds

The sources of noise from a Slocum glider have been well characterized (Moore, 2007). Flow noise was shown to possibly affect frequencies up to 2 kHz on a system that samples at 20 kHz. For the present work, flow noise was deemed not important since the principal interest was in high-frequency marine mammal sounds over 2 kHz. Other noise types made by the glider come from fin steering, movement of the battery, the volume piston, and the air pump. These are illustrated below, showing both their frequency content as well as typical time spans.

The fin acts as a rudder and controls the heading of the vehicle. Its typical noise signature as seen in the data, but also observed on the bench, is shown in Fig. 9. This is a very short duration noise of approximately 1 s or less with most of the frequency content below 5 kHz. It can be barely resolved in the example time series shown, whereas in both plots sperm whale clicks can be clearly seen.

The battery slides forward and backwards due to the pitch vernier mechanism, allowing the glider to descend and ascend, respectively. The volume piston pump moves water in an out of the glider's nose, which acts as a ballast compartment, to aid with descent and ascent. These actions occur concurrently when the glider reaches an inflection depth and either dives or ascends. Therefore, the noise associated with both battery movement and volume piston can be observed just prior to diving and just prior to ascending. An example observed in the recorded data is shown in Fig. 10. Because it happens at specific times during a dive, this roughly 20-second-long noise can be easily filtered out of the data.

The pitch pump, which moves the battery, can also come on for very short intervals during a dive to make small adjustments of the vehicle's pitch. The noise associated with this action is shown in Fig. 11. Finally, an air bladder, located in the rear section, helps raise the back end out of the water when at the surface so that the antenna can communicate with Iridium satellites. The air pump that inflates the bladder comes on only at the surface and should have no impact on the data recorded underwater and therefore is not shown here.

Because the glider's computer keeps track of the operations of all its motors and sensors, the exact time of the battery movement at an inflection point can be extracted. The glider clock is regularly updated through Global Positioning System (GPS) fixing when the vehicle is at the surface. This implies minimal clock drift. On the other hand, the Tascam clock is manually set prior to recording initialization. Therefore, glider self-noise can be an important feature to synchronize navigation and acoustic data. By looking at the time when the glider recorded a battery movement and change in battery position, just prior to an inflection depth, and comparing that with the acoustic recording of the battery noise, it was realized that the clocks were off by 76.61 s. This information was then used to synchronize both data sets (navigation and acoustics).

### 3.4    Glider Navigation Data

The glider diving profile during the recording of file 06 (used in the analyses presented here), is shown in the left plot of Fig. 12. While recording file 06, the glider performed roughly two complete descent-ascent cycles. It took roughly 16 minutes to descend to the maximum programmed depth of 170 m. On the other hand, it took almost twice the time, approximately 29 minutes, to climb back to 15 m. This difference indicates that the vehicle was not perfectly ballasted. Glider heading during this period, as measured by the vehicle and shown in the right plot of Fig. 12, was towards true north (0 degrees) with some oscillation. Pitch and roll were mostly constant during acquisition of the acoustic data (Fig. 12).

The glider was also fitted with a Sea Bird (SBE) pumped conductivity, temperature, and depth (CTD) sensor. From this data one can compute sound speed profiles representative of the survey area. Sound speed profiles can then be used as input to propagation models for characterizing the acoustic environment. Here the sound speed in sea water was calculated using the international standard algorithm (une) due to Chen and Millero (1977). The sound speed profiles calculated from data recorded by the glider are shown in Fig. 13.

### 4    The Acoustic Environment in the Survey Area

To characterize the acoustic environment where sperm whale clicks are propagated from some unknown location and recorded by the hydrophones fitted in the glider, the ray tracing model *Bellhop* (Porter and Bucker, 1987) was used to calculate incoherent transmission loss at the center frequency of sperm whale clicks (see Sec. 3.1). Transmission loss (TL) was calculated for different bearings by taking a fixed position for the glider along the track shown in Fig. 12. Here, the acoustic reciprocity principle (Kinsler et al., 1999) was used and calculations were made from a single point out to 20 km in range. It is noted that detection distances for sperm whale clicks have been reported in the literature between 5 and 16 km depending on environmental conditions and the propagation model used in the estimation.

Other input parameters assumed for TL calculations included the sound speed profile calculated from data collected by the glider and extrapolated to deeper waters based on the work of Salon et al. (2003). Three glider depths were assumed in the calculations: the minimum depth of a dive, or 15 m, the mid-depth of the dive at 80 m, and the maximum dive depth of 170 m. The bottom was assumed to be composed of sand with sound speed of 1700 m/s, density of 1.5 g/cm$^3$, and attenuation of 0.2 dB/m − kHz.

Results of propagation modeling at the center frequency of sperm whale regular clicks are shown as a function of range and depth for the bearing due north of the glider position, which is placed at the origin of the coordinate system (Fig. 14). Results for other bearings did not differ substantially. Given the greater depth of the bottom in relation to the glider's depth the seabed has little effect on propagation. From the three plots in Fig. 14 it can be assessed that detections were more likely to occur when the glider was closer to its maximum programmed diving depth of 170 m, where it is observed that most the water column is ensonified. In order to accurately predict detection distances, received levels (RL) must be known, the TL predicted and the source level estimated. For illustration purposes, assuming RL to be about 130 dB (see Fig. 3, for example) at the frequency of TL calculations, and an on-axis sperm whale click source level (SL) of 229 dB re 1 μPa rms (Møhl et al., 2003), would yield

(using the equation SL=RL-TL) a TL of 99 dB. Looking at TL curves as a function of range for a source at 500 m depth, that corresponds to distances of about 9 km if the glider was at 15 m, or about 12 km if the glider was at 80 or 170 m deep.

## 5    Marine Mammal Bearing Tracks

### 5.1    Bearing Estimation

In order to estimate bearing angles, automated detection of sperm whale regular clicks was performed by running a simple energy sum detector with the aid of the software *Ishmael* (Mellinger, 2001). Ishmael produces a detection function which represents the likelihood that a call of interest is present. The detection function has arbitrary amplitude units and a threshold is chosen with respect to its height (Mellinger, 2001). For this data set, a detection threshold of 0.05 was used to detect clicks with energy in the frequency band between 2 and 20 kHz, which is consistent with the frequency band of sperm whale regular

clicks (Zimmer et al., 2005). The energy sum detector was applied to channels 1 and 2 separately and detections were saved to corresponding files that logged initial and end times of each detection. Click durations from Ishmael detections ranged from 5 to 16 ms. Channel 1 produced more detections than channel 2 (43762 and 33325, respectively). Even though visual inspection seemed to indicate that glitches occurred more often on channel 2, their frequent presence on channel 1 could be a possible explanation for the larger number of detections. In addition, spectrogram levels were higher on channel 1 than on channel 2.

Therefore, some clicks detected on channel 1 probably did not have enough energy to be detected on channel 2. The cause for this difference in energy levels could not be properly assessed, but seemed to be connected to a malfunction of the hydrophone, which failed completely in a subsequent experiment.

Next, in order to estimate the direction from which the clicks came, the time difference of arrival (TDOA) of clicks received in the two channels was estimated. Due to the noisy character of the data, especially in the low and very high frequencies, a

bandpass filter was applied to the time series so that signals of interest could be distinguished. Hence, a fourth-order Butterworth bandpass filter was designed such that it had a flat frequency response between 1.5 and 25 kHz, with frequencies outside the 300 Hz to 43 kHz band attenuated 60 dB or more.

The TDOA can be estimated using various methods, the most common of which are cross-correlation and matched filter. Here, a biased estimate, which normalizes the cross-correlation by the number of samples, was calculated using the software

MATLAB. The correlation lag $\tau$, or time difference of arrival, is given by the maximum absolute peak of the cross-correlation of a time window containing a single detection. Instead of using cross-correlation between channels 1 and 2 to detect clicks and estimate $\tau$, we chose to use the detections provided by Ishmael on one of the channels. Here, channel 1 was used since it yielded more detections. A 6-ms window centered on each detection's initial time was extracted from channel 1 and cross-correlated with the same time window from channel 2. Such a time window was found sufficient to ensure that, in all observed cases, only one click was present in the time series. Longer detection windows provided by Ishmael were found to contain other multipath arrivals. Hence, by choosing a shorter cross-correlation window centered on a detection's initial time, it minimized (or eliminated) errant correlations between direct and multipath arrivals.

By assuming a nominal sound speed of 1500 m/s in the ocean and taking the hydrophone separation of 0.9 m, it was found
that the maximum possible TDOA between arrivals of a click on both hydrophones was $T = 0.6$ ms. This provided a means of
rejecting glitches or other false positive detections on a single channel. It is noted that the sampling frequency with which the
data were recorded provided good time resolution ($\Delta t = 0.01$ ms) relative to $T$.

The estimated TDOA was then used in the formula below to find the direction of arrival of each detected click, keeping in
mind the inherent left-right ambiguity of the estimate. The direction of arrival, or bearing angle $\theta$, was calculated by

$$\theta = \cos^{-1}\left(\frac{\tau c}{L}\right), \tag{1}$$

where $c$ is the sound speed (1500 m/s) and $L$ is the hydrophone separation distance (0.9 m). Bearings are estimated between
$0°$ and $180°$ (with ambiguity from $0°$ to $-180°$). Therefore, the final step was to convert $\theta$ to angles in the glider's reference
frame ($0°$ to $360°$). Estimated bearings can not be readily corrected for the glider's recorded heading due to the ambiguity of
the estimates. Results and related accuracy of the bearing estimates are presented next.

## 5.2 Bearing Results and Identifying the Left-Right Ambiguity

Clicks (sperm whales) present in just over one minute of data from the onset of file 06 were manually annotated and compared
to detections made by Ishmael as a qualitative measure of detector performance. The results of this comparison given in terms
of bearing angles (not corrected for glider's reference frame) calculated from both sets of detections are shown in Fig. 15. For
this short period of time, manual annotation yielded 399 clicks (or bearings) while the automatic detector produced 406 clicks.
Of the 406 automatic detections, 84 were in fact false positives, corresponding to about 21% of the detections in just over one
minute of data. Both detection methods agreed relatively well, and this preliminary result suggests that at least two animals,
possibly three, were producing echolocation clicks during this time of the recordings.

Bearing angles estimated for all true detections of file 06 from the cross-correlation analysis are shown in Fig. 16. This
corresponds to just over one hour and thirty minutes of data. The bearings were corrected for the glider's reference frame and
both glider heading and dive profile are plotted on the same figure for reference. Glider heading in Fig. 16 (the same as in Fig.
12) was shifted by $50°$ for plotting purposes. Note, however, that estimated bearings given in Fig. 16, are not necessarily the
correct bearings. A different set of angles, opposite to the ones plotted and which correspond to the left-right ambiguity are also
possible solutions. Each bearing angle corresponds to a click detection. Thus, observing detections along the glider track does
not indicate, on first glance, a preferred depth where detections occur as suggested by the TL plots shown in Fig. 14. Four small
gaps in detection can be observed in Fig. 16, due to the movement of the battery when the glider starts a descent or ascent. The
color scheme corresponds to the strength, or peak, of the cross-correlation in decibels (dB). The weaker the cross-correlation
peak, the harder it is to differentiate the click above the noise floor. It is noted that sperm whale vocal activity was observed
through out file 06, whereas dolphin clicks seemed to be present mostly in the last 20 minutes of data. Shorter time segments
within this figure are examined in detail to get a better picture of the tracks.

Three shorter segments of estimated bearing angles are shown in Fig. 17 (a)-(c). The upper plot (Fig. 17 (a)) shows bearing
angles estimated from clicks recorded during the first 8 min from the beginning of file 06. By zooming into this shorter period

of data, it is possible to realize three different tracks closely following each other. The glider heading is also observed to nicely follow the bearing track for this set of angles, as opposed to their ambiguous counterpart (not shown here). The second zoomed in plot (Fig. 17 (b)) shows 18 min of estimated bearings in the middle of the file. One strong track is seen throughout, following an opposite pattern as the glider heading. This feature was not found to be related to multipath clicks (Sec. 3.1), which were not picked by the detector. When the detector did pick multipath clicks they were part of the same detection as the first arrival and hence, could be excluded from the cross-correlation process. Therefore, multipath clicks did not have any significant contribution to the bearing results presented here. Two other tracks are also observed in the same plot, following closely the glider's heading, as before. Finally, the last plot, Fig. 17 (c), shows just over 20 minutes of bearings estimated at the end of the file, where dolphin clicks were more predominant. It is worth noting that detections were made between 2 and 20 kHz; hence, estimated bearings in this window could correspond to either sperm whales (the target of the click detector) or dolphins, whose clicks had enough energy in the sperm whale frequency band to elicit a detection. At first glance, the results shown in this plot seem to indicate the presence of a few tracks, with strong cross-correlation peaks between 85 and 90 minutes. These stronger correlations seem to almost form a different and separate set of tracks.

Finally, a polar plot (Fig. 18) was made combining all estimated bearings. This shows the clicks' directions of arrival in the glider's reference frame, including the left-right ambiguity inherent in a two-sensor arrangement. A diagram of the glider depicting the left-right ambiguity is also shown for illustration purposes. The polar plot suggests that most clicks were coming from approximately northeast, or southeast, of the glider, which was heading approximately north during the recording of the data. A second, smaller group of clicks coming from the northwest, or southwest, also seemed to be present.

## 5.3 Bearing Accuracy

The accuracy of bearing estimates depends on different factors such as the time-difference of arrival (TDOA) estimation via cross-correlation of data recorded on the two channels, the signal-to-noise ratio (SNR) of received signals, and the accuracy of the automated detection process. Accuracy can also be thought of in terms of the ability to resolve or distinguish two sound sources that are close to each other. The accuracy of the detection algorithm was shown qualitatively in Sec. 5.2, where manual and automatic detections were compared for just over a minute of data, and yielded good agreement.

To assess the accuracy of the cross-correlation process a snippet of 100 ms of data containing noise and one sperm whale echolocation click with good SNR was randomly chosen and extracted from both channels for analysis. The time delay between the two channels, estimated by cross-correlation, was 0.3125 ms. Next, a smaller time window containing only noise was further extracted from the 100 ms data snippet. The click signal was extracted from channel 1 only. Two waveforms were then created: one was the combination of the extracted noise and click from channel 1, and the other combined the noise from channel 2 with the click from channel 1, which was time delayed by 0.3125 ms. Cross-correlation of these two waveforms, with known time delay, yielded an estimated lag of 0.3229 ms. Translating the lags to bearings using Eq. 1 yields 121.4° and 122.6°, respectively. The error in time delay estimation by using cross-correlation corresponded to a difference of approximately 1.2° in bearing.

The effect of click SNR on estimated bearings was also examined by using the two waveforms created as described above. Signal-to-noise ratio was measured from computed spectrograms of the waveforms. Spectrograms were calculated by using a fast Fourier transform (FFT) with 1024 points, Hamming window, and 50% overlap. The power of both noise and signal were summed between 2 and 20 kHz. The decibel (dB) values of those quantities were then subtracted, yielding SNR values in decibels. The click, extracted from channel one and used to create the short waveforms, had a measured SNR of 9.4 dB. Noise power levels were then modified and both click SNR and time delay (bearing) were estimated. The result is shown in Fig. 19 as a plot of SNR versus bearing angle. By decreasing noise levels and consequently increasing click SNR, a small decrease was observed in the estimated bearing angle of approximately $1.2°$. However, it does not matter by how much noise levels are decreased (or how high SNR is), the change in bearing is constant. On the other hand, increasing the noise levels (i.e., decreasing click SNR) lowered the estimated bearings even more. An SNR decrease of 1 dB corresponded to a difference of $3.5°$ in bearing angle. Furthermore, dropping the SNR by 3.4 dB caused a decrease in estimated bearing from $122.6°$ to $88°$, or a difference of $34.6°$. It is noted that these results corresponded to only one click sample from the data set.

## 6    Discussion and Conclusions

In this work, a glider was fitted with two hydrophones and an inexpensive, off-the-shelf acoustic recording system for use in studies related to marine mammal population density estimation. Even though the experiment described here was opportunistic and by no means designed as a density estimation experiment, this was the first time a glider fitted with two sensors was used to monitor marine mammals. Evaluation of glider operations and of the acoustic system was performed during the REP14-MED sea-trial off the west coast of the island of Sardinia, Mediterranean Sea. About 20 hours of dual-channel continuous acoustic data were recorded in deep water (>2000 m), and contained calls of sperm whales as well as dolphins. Sperm whale regular clicks recorded on both channels were cross-correlated for the estimation of bearing angles, and animal tracks could be recognized from this analysis. Only a few studies exist on the distribution and abundance of sperm whales in the Mediterranean Sea. The current work contributes not only with a unique data set from which sperm whale tracks could be realizable but it also adds to the pool of information of where such animals might occur. In terms of density estimation studies, the acoustic data recorded by the glider provides a good starting point for extending the existing methodology to slow moving platforms. More specifically, the ability to estimate animal tracks from estimated bearing angles provides a distance-related co-variate that has been shown to increase accuracy of density estimates in a terrestrial study.

The successful use of a good quality and inexpensive voice recorder connected to a pair of hydrophones led to subsequent improvements to the system. In its original configuration, the Tascam recorder did not allow for the implementation of any recording schedule other than continuous recording, restricting data collection to a maximum of 23 hours. On the other hand, Slocum gliders have the potential to stay deployed for a few weeks at a time. Another drawback of the recording system was an inability to start and stop recording via remote command. Thus nearly two hours of data, almost 10% of total capacity, were recorded while the vehicle was still on board the NRV *Alliance*. An improved second generation was devised after this experiment with added storage capacity and connected to a micro-controller serving as a programmable interface.

The quality of the data was generally acceptable, and even though recordings amounted to less than a day, sperm whales and dolphin calls were identified over several hours in the data set. However, random short-duration glitches of seemingly electronic origin were also present throughout, but not concurrent on, both channels. Some investigation has linked the source of such glitches to a defect in the circuit board that powered the hydrophones and connected these to the Tascam recorder. Even though some processing needed to be done in order to remove glitches from detections, they did not compromise the usability of the data set. Both hydrophones were from the same manufacturer, with the same sensitivity and pre-amps, but their outer shells were slightly different. On a more recent experiment one of these hydrophones stopped working completely and it came to our attention that water might have leaked inside the sensor. This could potentially explain the difference in levels observed between the two channels.

Detection of thousands of sperm whale and dolphin clicks in a data segment of approximately 1 hour and 30 minutes was enough to test the usefulness of two hydrophones in the glider for marine mammal population density estimation studies. Some advantages of having two sensors mounted on a glider, instead of a single hydrophone, include: 1) bearing angle estimation, which can be used as an additional co-variate in density estimation methods, thus increasing the accuracy of estimates (e.g. Glennie et al., 2015); 2) the potential to estimate animal tracks, which can give another measure of how many animals are present in a given location surveyed by the glider; and 3) estimated tracks can also be used to infer inter-call intervals, which are an important parameter necessary for estimating the percentage of time a species produces sound during one day (Marques et al., 2013). Effects of glider movement, especially displacement with depth, will mostly impact detection functions, which are used to estimate the average probability of detecting calls, another parameter important to density estimation. Estimating the glider detection function is a current and on-going topic of research by different groups that are using gliders for population density estimation studies.

Looking at the track results (Figs. 16 and 17) and their relation to the glider heading, it may be possible to disambiguate identified tracks by using the observed oscillation in the glider heading. As observed in Figs. 16 and 17, some estimated bearing tracks followed closely the glider heading, whereas a couple tracks clearly had an opposite pattern likely due to the incorrect assumption about which side of the hydrophone the sounds were coming (the left-right ambiguity). Another interesting question that needs to be answered is the resolution of the tracks and the angular separation needed to distinguish one animal from another. More data analysis needs to be done to identify the dolphin species observed in file 06. If enough energy from dolphins' clicks are present in the data, their tracks can be potentially resolved. Longer tracks could also be realized by combining results from multiple files and observing the continuation of clicking activity.

Finally, a major hardware malfunction was identified in the glider during the sea-trial. A corrupt piece of hardware affected the its navigation and communications. Fortunately, the problem was tracked down with the help of the engineers on board the NRV *Alliance* after Clyde was recovered. A new piece of hardware was subsequently installed and glider operations have resumed normally.

*Acknowledgements.* The authors would like to thank Eric Sorensen for the invaluable help in adapting the Tascam recorder for use with the NEAR-Lab glider. The authors would also like to thank the scientists, engineers and participants of the REP14-MED, the crews of NRV

Alliance and RV Planet, and NATO Science Technology Organization, Centre for Maritime Research and Experimentation (STO-CMRE) for their help in testing, preparing, deploying and recovering Clyde. Special thanks are in order to Reiner Onken (the scientist in charge on NRV Alliance), Richard Stoner (engineering coordinator), Rod Dymond (acoustics engineer), and Bartolomeo Garau (glider pilot/operations) without whom this data collection would not have been possible. The authors would also like to acknowledge the Office of Naval Research (ONR) Marine Mammal and Biology Program for funding the project that led to this research. This is PMEL contribution number 4524.

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

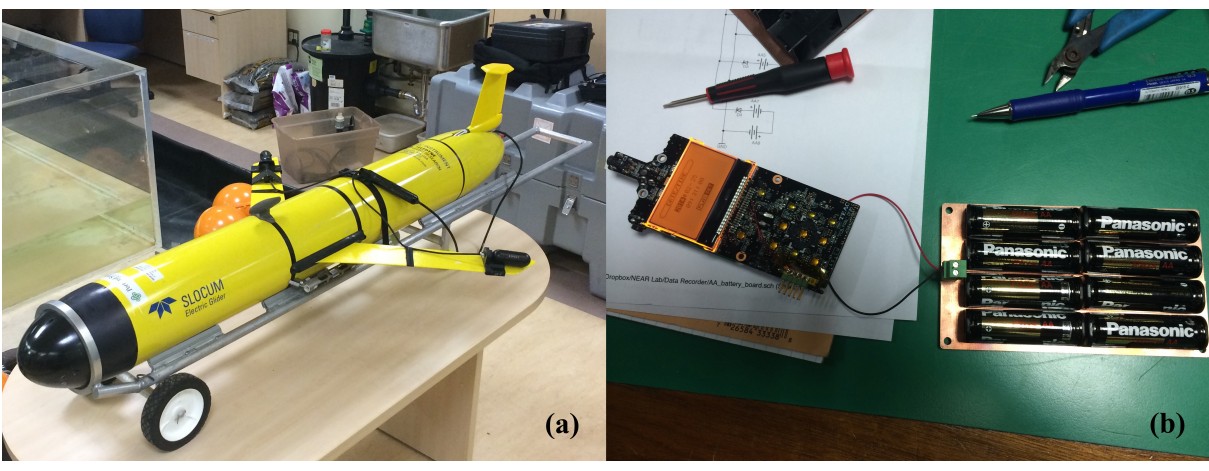

**Figure 1.** (a) The PSU glider *Clyde* with hydrophones attached to the tips of its wings. (b) The acoustic recording system (modified Tascam digital recorder) and battery pack.

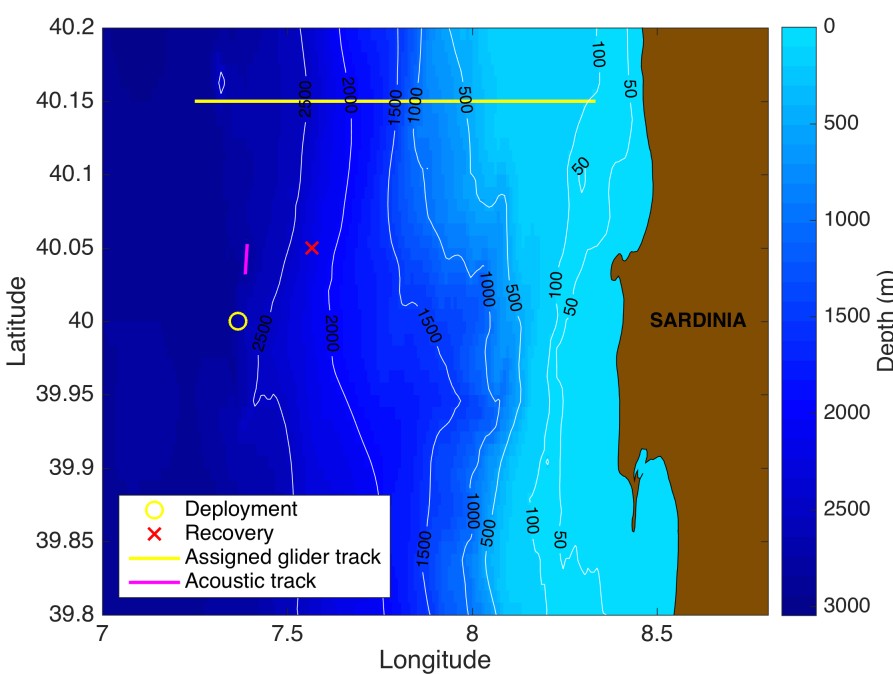

**Figure 2.** Bathymetry off the west coast of the Island of Sardinia, Italy, showing the glider's deployment (yellow circle) and recovery (red cross) locations, as well as the glider's pre-assigned track for the experiment, and the glider trajectory (vertical magenta line) during the recording of file 6, used in this work. Due to a failure in hardware, the glider never made it to its assigned track. Note that the glider flew for approximately one day from the deployment day, but was recovered 3 days later, having drifted east from its original trajectory due north.

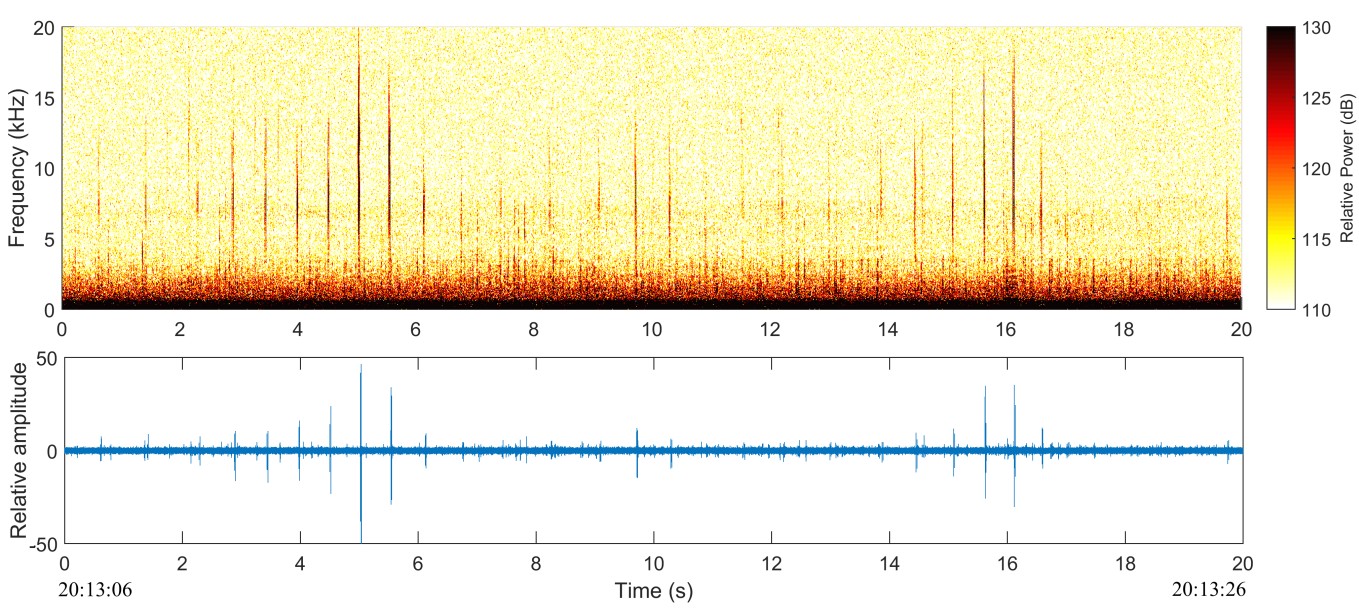

**Figure 3.** Spectrogram (top) and waveform (bottom) of 20 seconds of data recorded on channel 1 of file 06 showing sperm whale regular clicks (narrow vertical bars). Time stamps are local time (CEST) on 09 June 2014. The relative amplitude corresponds to the amplitude in dB minus the hydrophone sensitivity of -161.4 dB re 1 μPa/V.

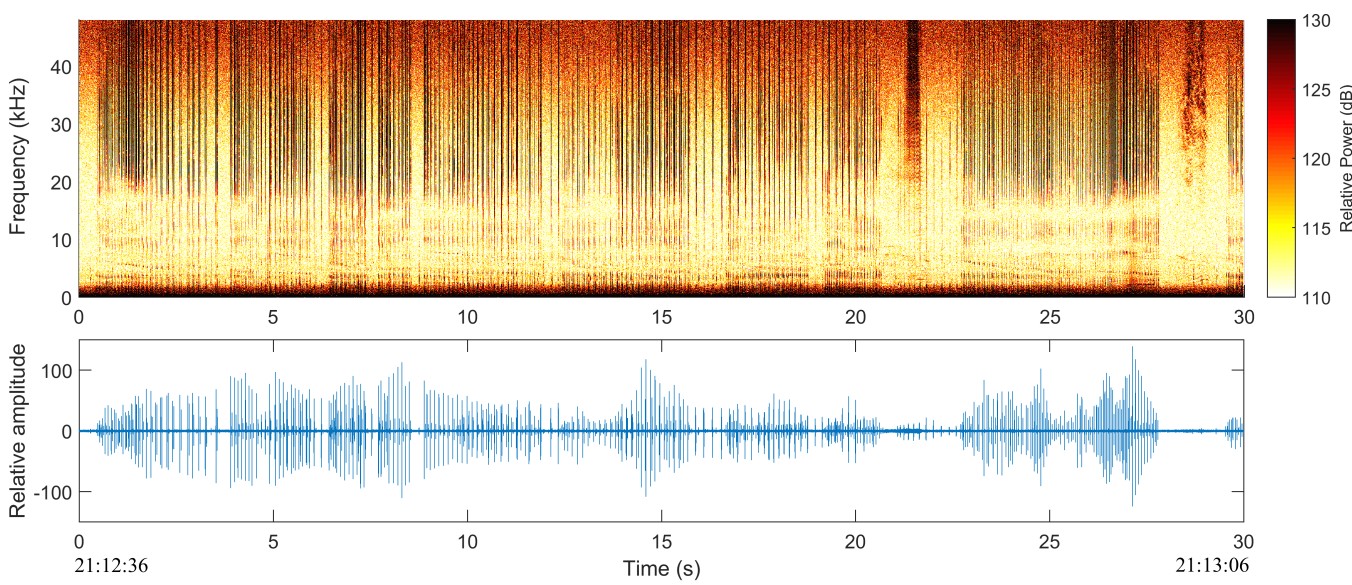

**Figure 4.** Spectrogram (top) and waveform (bottom) of 30 seconds of data recorded on channel 1 of file 06 showing dolphin clicks (vertical bars mostly above 15 kHz), and burst pulses (mostly above 15 kHz between 20-22 s and 28-30 s). Note that the frequency range is different from the previous plot. Time stamps are local time (CEST) on 09 June 2014. The relative amplitude corresponds to the power in dB minus the hydrophone sensitivity of -161.4 dB re 1 μPa/V.

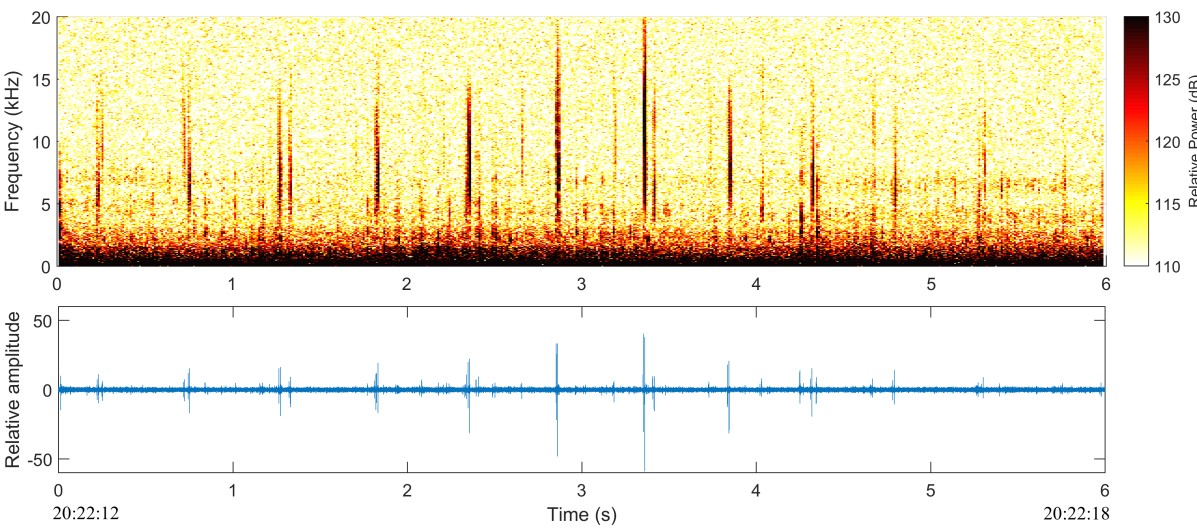

**Figure 5.** Example of possible sperm whale multipath arrivals - here evidenced by fainter clicks following the stronger first arrivals, more noticeable between 2-5 s and shown on both spectrogram (top) and waveform (bottom) of 6 seconds of data from channel 1.

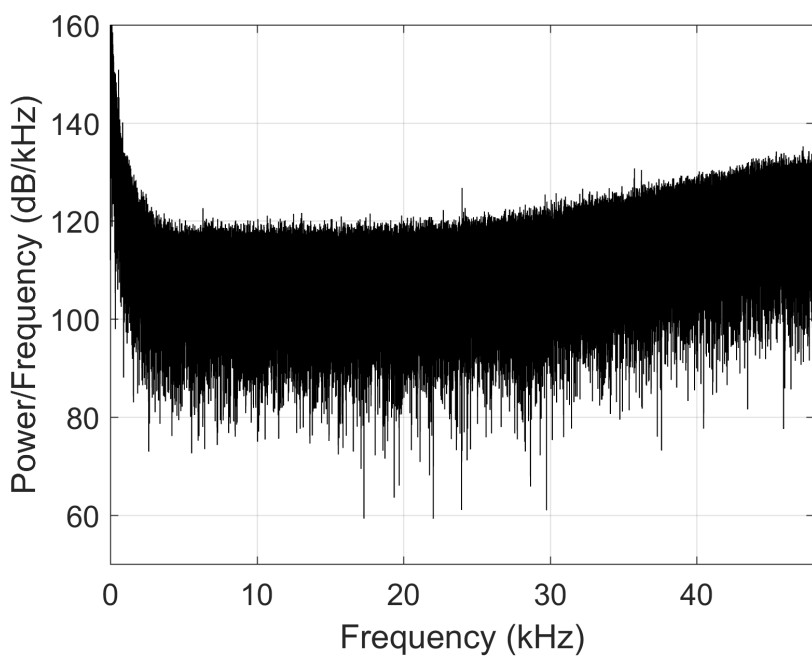

**Figure 6.** Power spectral density estimated using Welch's method from 5 seconds of data containing only background noise, showing the increase of power for frequencies above 25 kHz.

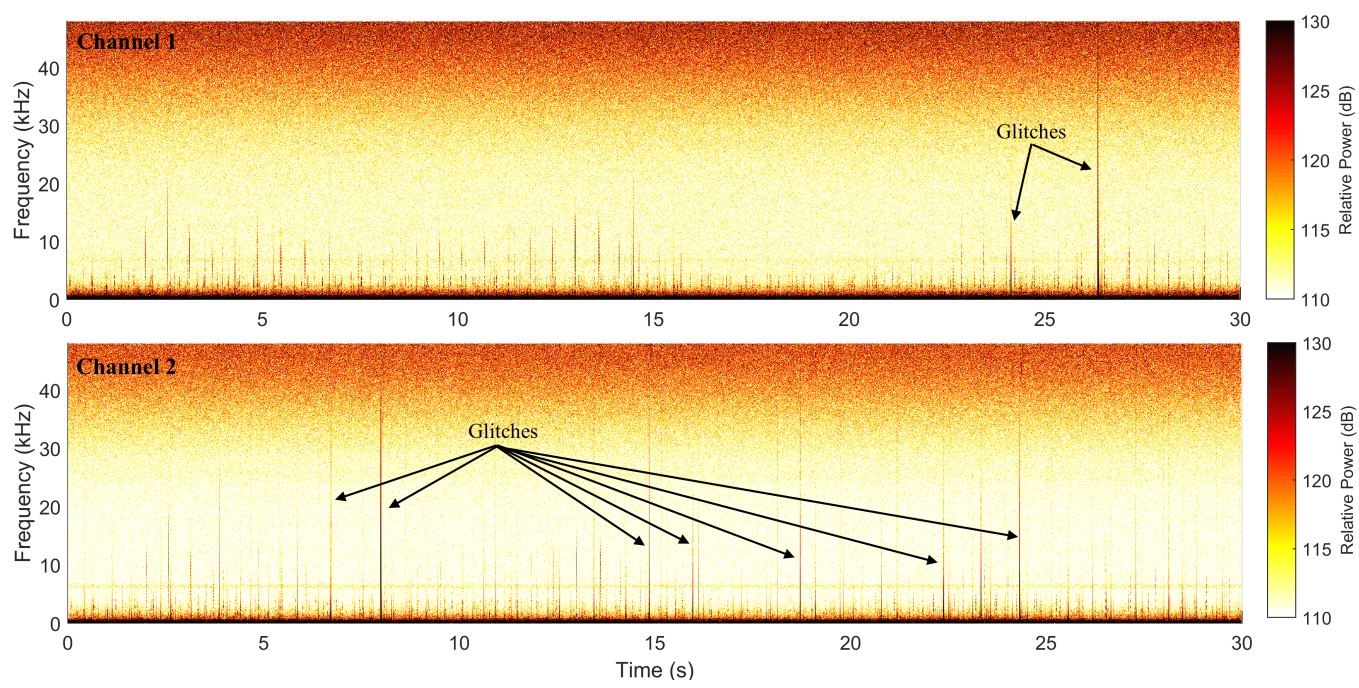

**Figure 7.** Spectrogram of 30 seconds of data recorded on channel 1 (top) and channel 2 (bottom) showing the occurrence of glitches for the same period. While there were only two instances when glitches occurred on channel 1, they showed up many times on channel 2 (only a few are actually shown).

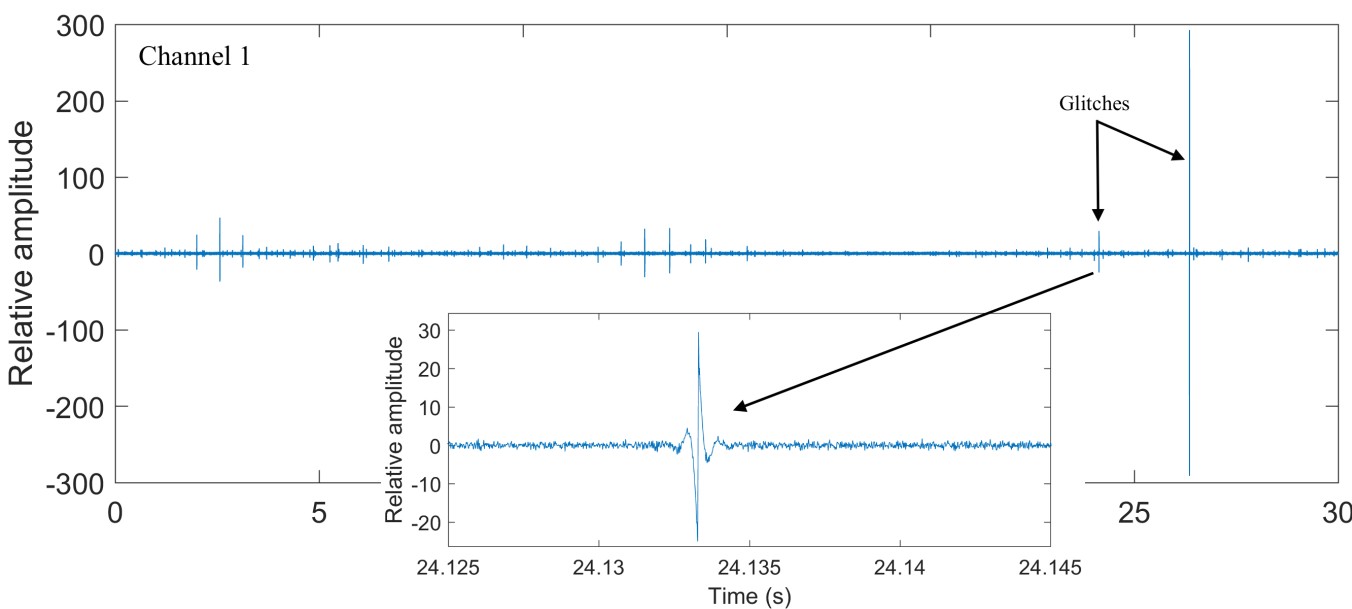

**Figure 8.** Time series recorded on channel 1 for the same 30 seconds of data shown in Fig. 7. The two glitches present are indicated by arrows. The inset shows detail of one of the glitches. Regardless of the amplitude, which can be comparable to the amplitude of odontocete echolocation clicks, they all have the characteristic shape shown here, which is distinctly different from that of echolocation clicks.

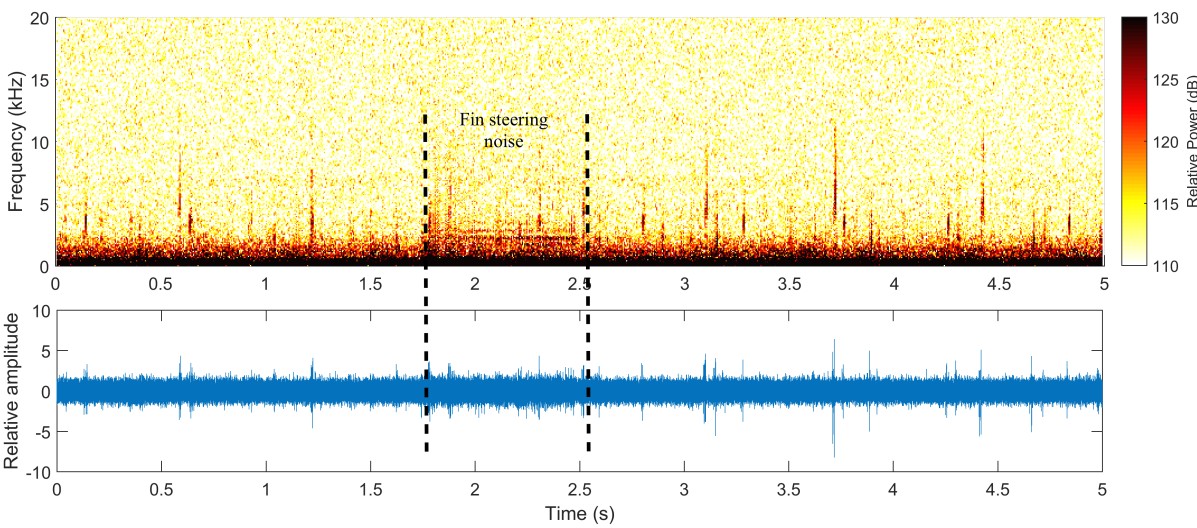

**Figure 9.** Spectrogram (top) and time series (bottom) of 5 seconds of data showing an example of noise (between dashed lines) produced by fin steering in the glider, with sperm whale echolocation clicks (narrow vertical bars) around it.

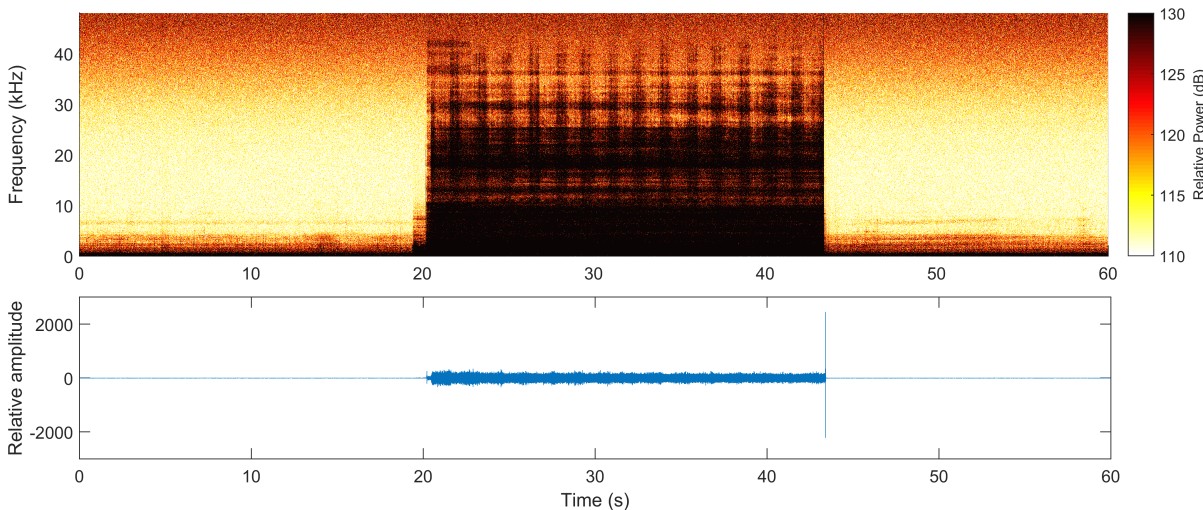

**Figure 10.** Spectrogram (top) and time series (bottom) showing an example of the noise produced by battery movement and volume piston. The volume piston noise starts just before 20 s, and is then masked by the battery movement noise.

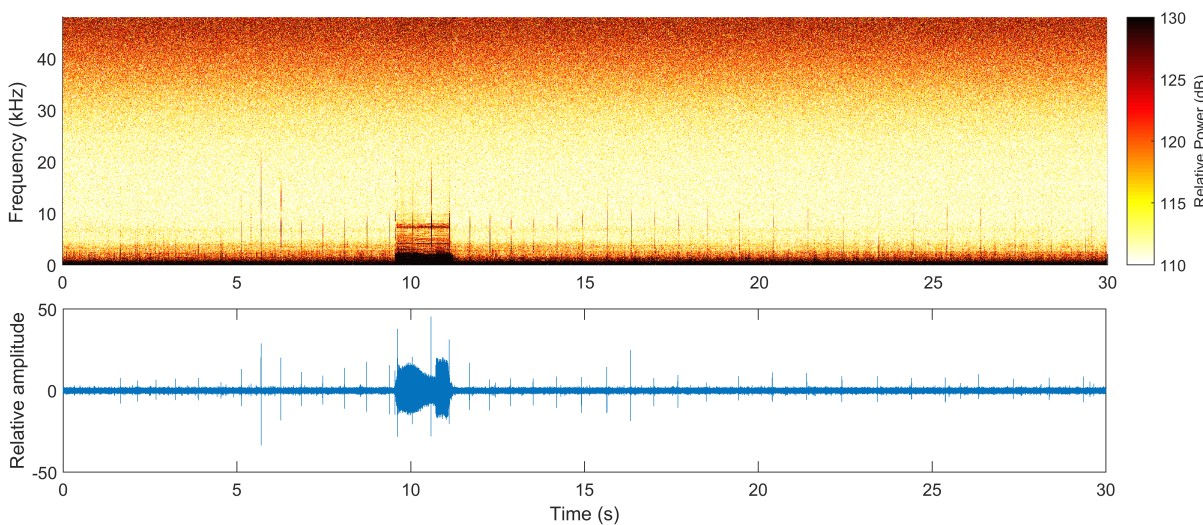

**Figure 11.** Spectrogram (top) and time series (bottom) showing an example of noise produced by the pitch pump, which makes small adjustments to the battery position during a dive. Sperm whale echolocation clicks can also be observed in this sequence, especially in the time series data.

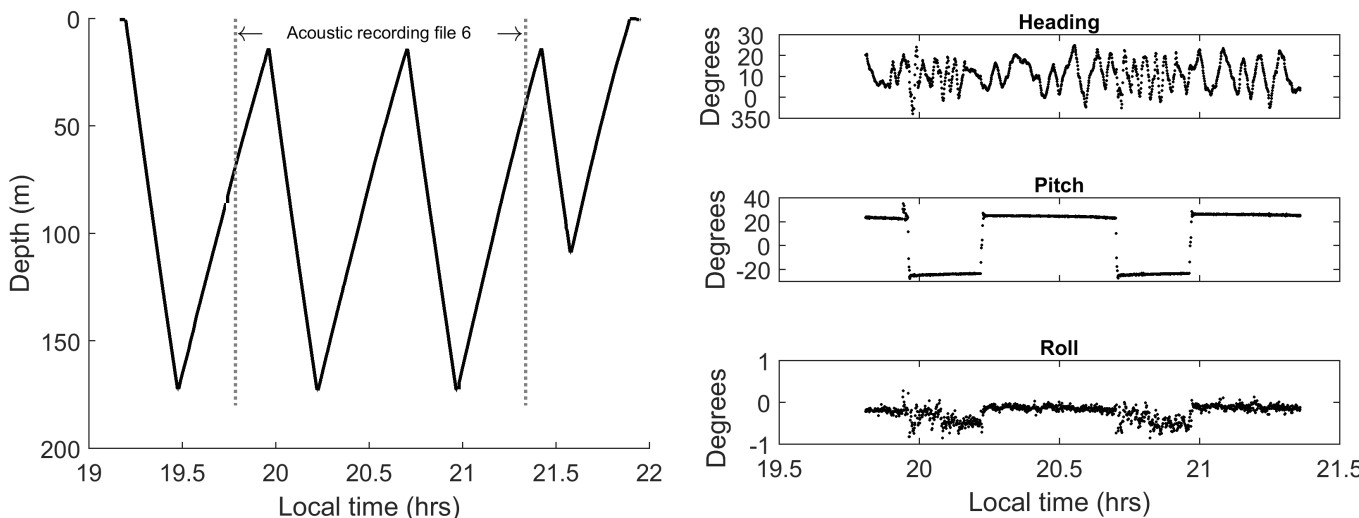

**Figure 12.** The left plot shows the glider's dive profile during the recording of the acoustic data used in this work. Navigation parameters of the glider (heading, pitch and roll) are shown on the right plot. Note the highly oscillatory pattern of the glider heading.

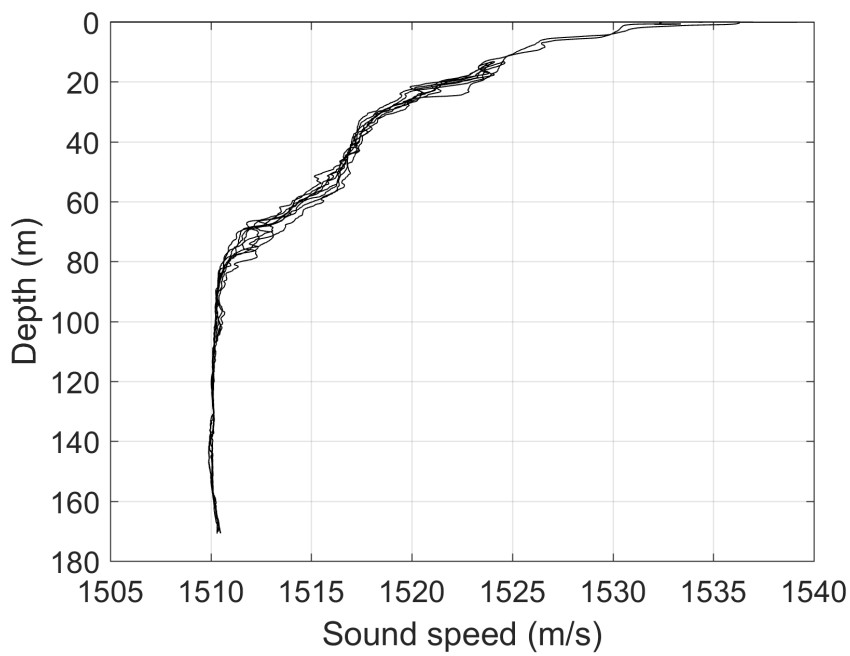

**Figure 13.** Sound speed profiles calculated using conductivity, temperature and depth measurements made by the glider during the recording of the acoustic data.

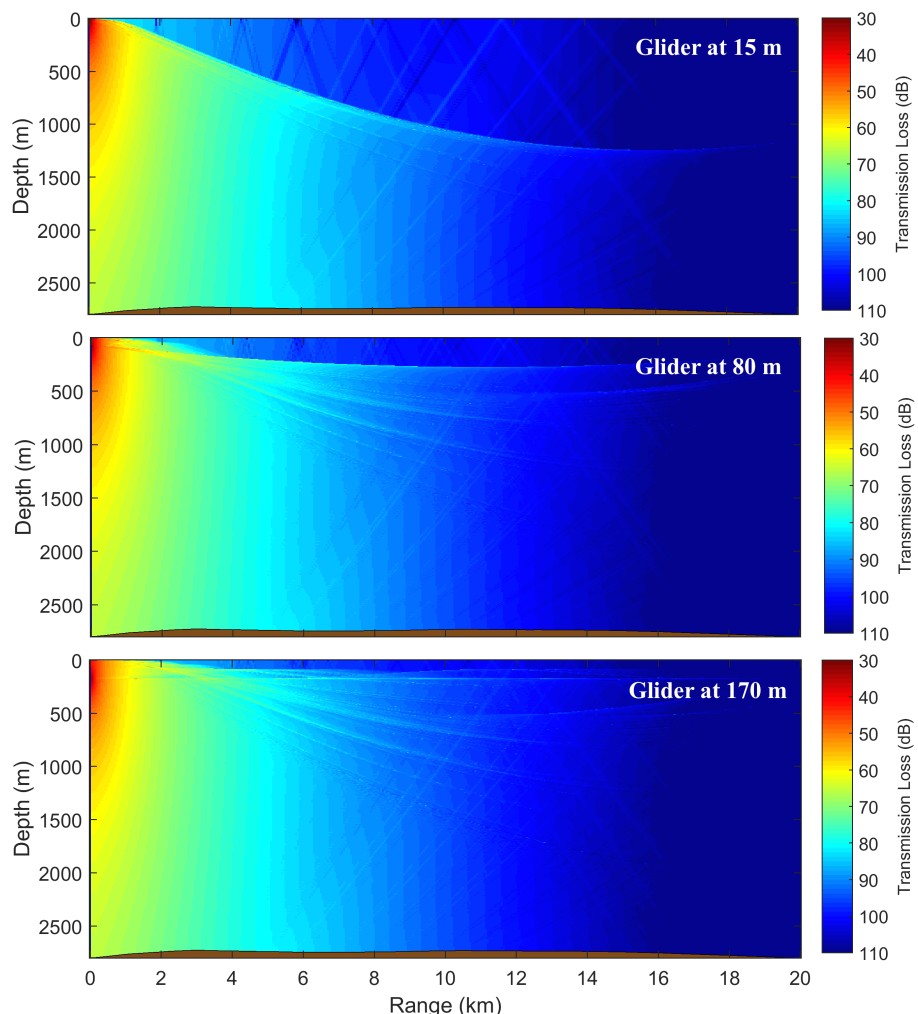

**Figure 14.** Transmission loss as a function of range and depth for 3 glider depths (15, 80, and 170 m), and for a source frequency of 13.4 kHz. The bearing of the plots is due west of the glider position at 40° 2.6' N 07° 23.45' E. As the glider moves deeper, the surface shadow zone narrows and caustics (regions of high intensity) appear.

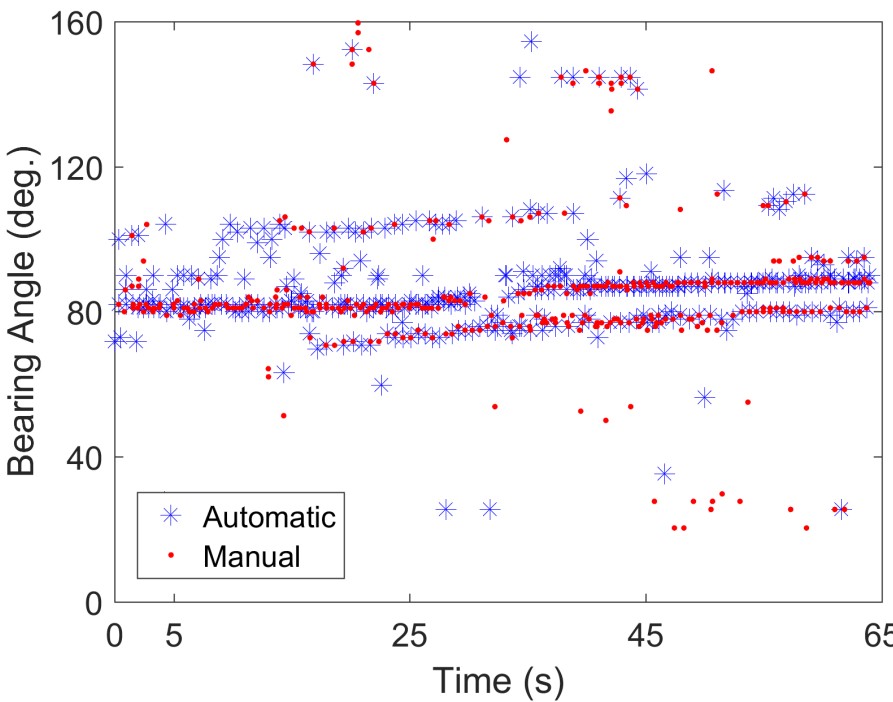

**Figure 15.** Estimated bearing angles from automatic (blue stars) and manual (red dots) detections made just after the beginning of file 06.

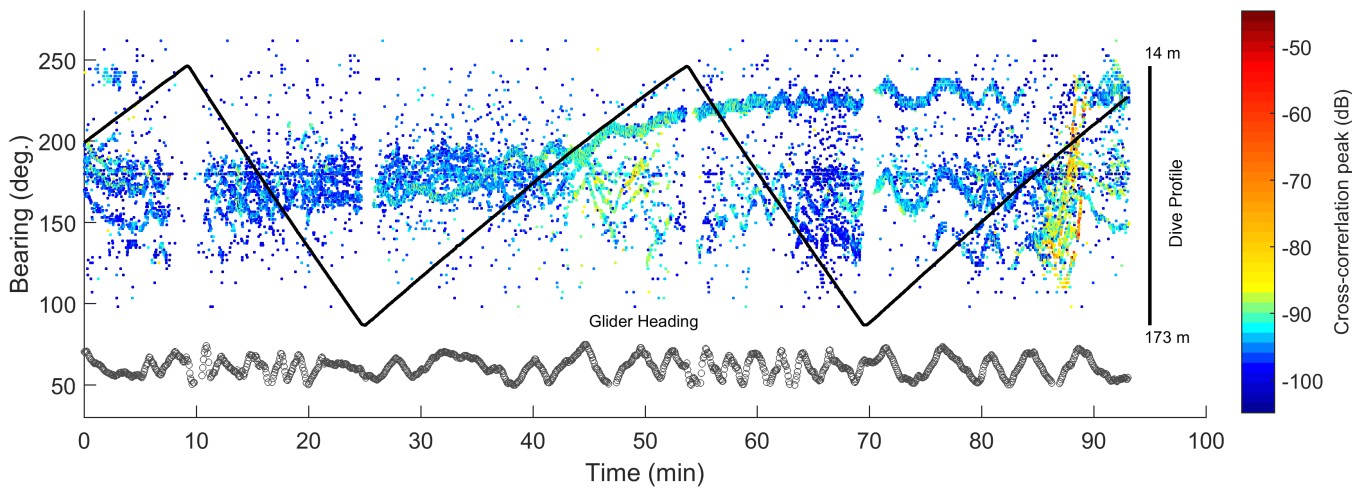

**Figure 16.** Estimated bearing angles (ambiguous angles are not shown) relative to the glider of all clicks detected in file 06 as a function of time and cross-correlation peak strength (colorbar). The glider's heading is shown below (shifted up by 50°), and the glider's dive profile (black zigzag) is superimposed for reference.

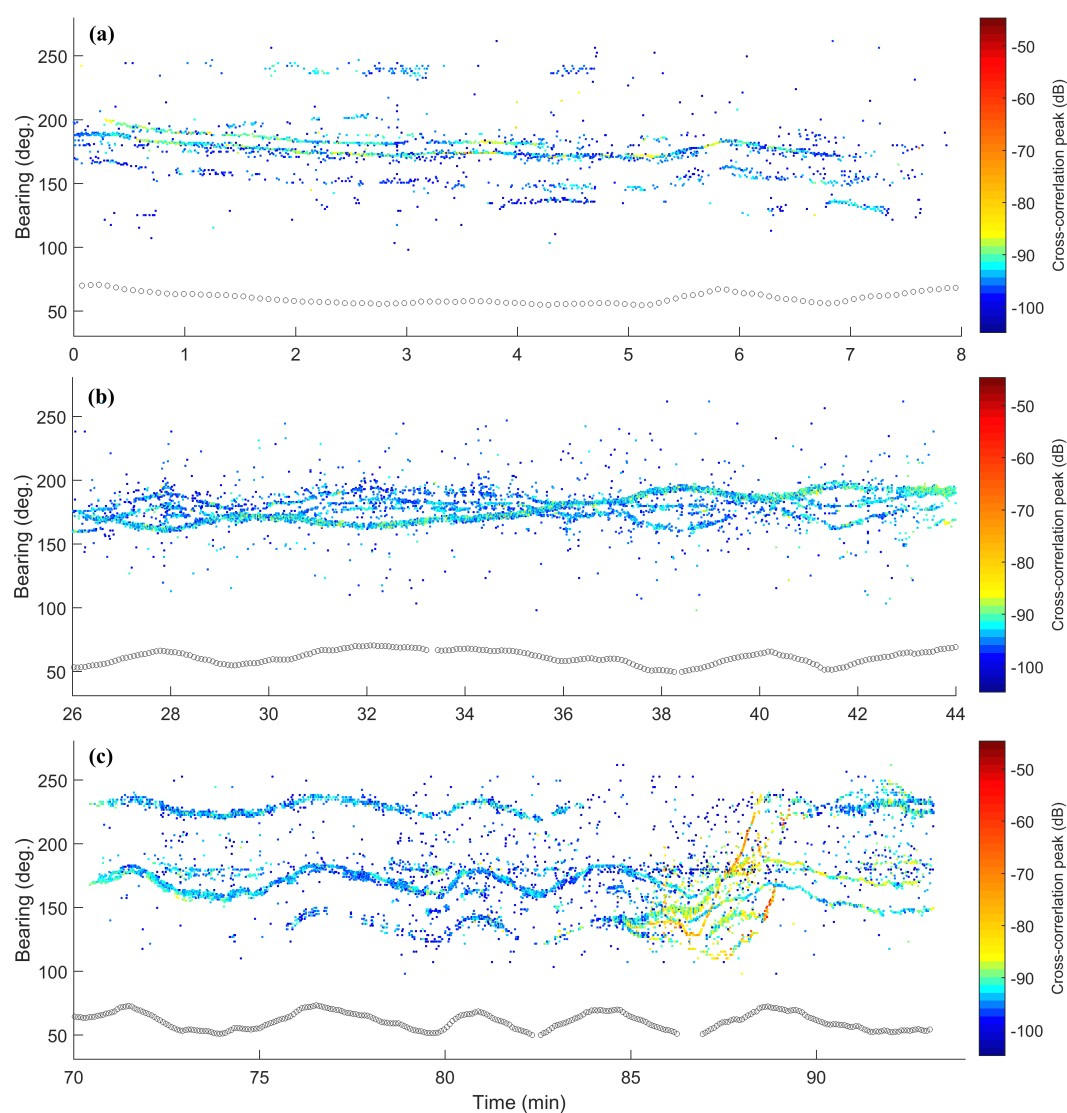

**Figure 17.** Estimated bearing angles from automatic detections of file 06, for three smaller time windows: (a) 0-8 minutes, (b) 26-44 minutes, and (c) 70-95 minutes. The glider's heading is shown as the line of gray circles below.

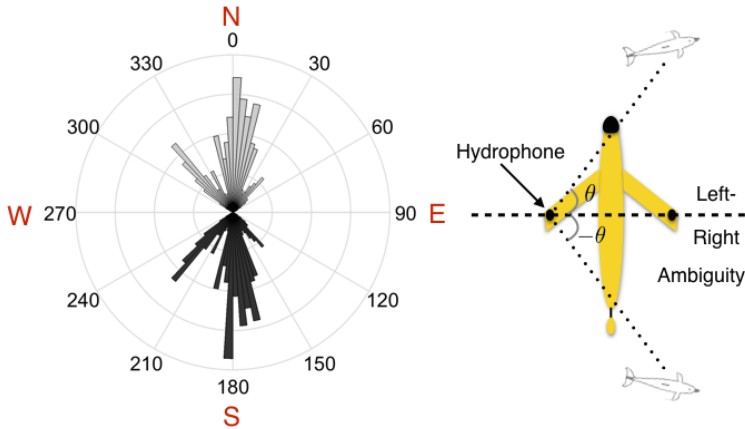

**Figure 18.** Estimated bearing angles from automatic detections of file 06 shown in polar form. Angles are relative to the axis of the glider, which was heading north (0 degrees) during data recording. Because the hydrophones were mounted horizontally, the bearings have front/back ambiguity, shown in the diagram on the right.

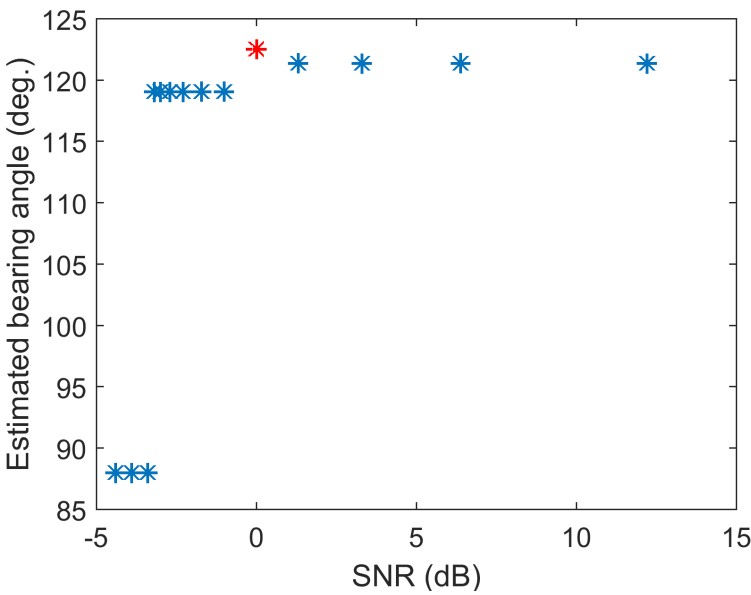

**Figure 19.** Click SNR plotted as a function of estimated bearing angle. SNRs, computed by increasing and decreasing noise levels, are given relative to the original click SNR (plotted as a red star), as measured from the data set.