# Peer review of "Marine mammal tracks from two-hydrophone acoustic recordings made with a glider"

_Ocean Science, 2016_

## Referee Comment (RC1) · Anonymous Referee #1 · 2 Jul 2016

the paper is interesting as it offers a report on the use of gliders for performing acoustic surveys to detect and study marine mammals. The specific case present and interesting option based on a low cost recorder rather than custom complex dedicated electronics. However the paper appears more as a basic tech report than a scientific paper. The findings have no scientific relevance for marine biology and the authors show little expertise in the description of detected biological sounds. Dolphin clicks and sperm whale clicks are well known now. The figures don't show the characteristics of detected events in detail, e.g. to clearly show the differences among artifacts and real signals, or to show the multi paths underlined in the text. The multi paths in recording biosonar clicks is well known and the multi paths can be positively used to improve the localization od sperm whales. Surface multi paths are generate by the sea surface, but often also the sea bottom generates reflections of sperm whale clicks. With a flat

sea surface reflected clicks show phase inversion, described in the text as mirror images. Advantages/disadvantages of the use of a glider are not presented. Which is the impact of flow noise ? How the change in depth influences the recording ? Which type of noises are made by the glider itself, e.g. when it changes its asset ? is the quality of the recorder well suited to the task ? Authors write about clicks with energy content increasing with frequency. Most dolphins do produce clicks with peaks above 40 kHz and up to 100 kHz and more. Recording them at close range may result in very high frequency levels that may saturate the hydrophone, its preamplifiers and even the recorder input. Also to consider the resonance of the ceramics in the hydrophones and the possible aliasing effect induced by the intrinsic a-a filters of the recorder that may "reflect" the acoustic energy above Nyquist down to the recorded range. A minor point concerns the choice of the recorder. External batteries have been used. Other pocket recorders have less noise and require much less power than the Tascam. Some can run for 48 hours on their two internal AA batteries. The recorder is called "voice recorder" but it should be called "music recorder".

---

## Referee Comment (RC2) · Anonymous Referee #2 · 12 Sep 2016

Review of "Marine mammal tracks from two-hydrophone acoustic recordings made with a glider," by Elizabeth T. Küsel, Tessa Munoz, Martin Siderius, David K. Mellinger, and Sara Heimlich

This manuscript is a descriptive report of the performance of a Slokum glider with two hydrophones mounted on the wings of the glider, providing an 0.9 m spacing between hydrophones. A modified commercial TASCAM stereo recorder with about 23 hours of operational time (limited by the 32 Gbyte of memory storage at 96 kHz sampling rate).

This manuscript is worth while publishing only because the use of acoustics on glider for marine mammal detection is in its infancy and it's important to share various investigators' experiences and results from their field test. In this case, 23 hours of recordings were achieved but most of the data occurred in a 1 hour time span. However, I do have

a number of misgiving about this manuscript and are basically involve the avoidance of serious discussion about the usefulness and accuracy of the results. I will details some of the items that the authors should address in a revision.

1. The accuracy of the bearing estimates is never discuss and I think it needs to since I believe the accuracy was not very high. The baseline is too short and the further out the animals are from the glider the more inaccurate the estimate. Also the position of the animals with respect to the glider direction will have a big effect on the accuracy. The dynamics of the glider, especially the yaw, is not even mentioned. The localization is discussed in a manner that suggest no problems, not issues, perfect localization. I think this issue is considerably more important than the techniques used for localization since time of arrival difference based cross correlation analysis is fairly routine.

2. There is some hand waving in the statement "Such information can be valuable to density estimation methods, either directly for estimating the percentage of time a species produces sound during one day (Marques et al., 2013)." If you have a moving platform and come across a group of animals also moving, directly estimating the percentage of time a species produce sounds can surely be done but what does it mean? How such (bearing estimate) information be valuable to density estimation methods seems like a good statement to make but is it really true with poor bearing accuracy?

3. There should be a better way of displaying click signals then a spectrogram. All you see is a line going to very high frequency (off the chart in some cases) and that's support to tell me more than the time of occurence? How's about plotting center frequency or peak frequency instead?

4. A minor issue is the phrase in the last line of page 3, ". . ..where high frequencies are highly attenuated." I don't know what highly attenuated means? At 30 kHz the absorption coefficient is about 3.9 dB/km and at 15 kHz its about 1.0 dB/km. I don't consider the 2.9 dB/km difference very large in the broader scheme of ocean propagation.

5. I don't understand why click ID software such as M3R is not used to try to ID some

of the deep diving odontocetes like beaked whales, Risso's dolphins and pilot whales.

---

## Author Response (AR1)

Dear Topic Editor,

We are grateful for the insightful comments from the anonymous referees. What follows is our detailed point-by-point response to their comments on the manuscript "*Marine mammal tracks from two-hydrophone acoustic recordings made with a glider,*" by Elizabeth T. Küsel et al. This report also points to the changes made to the original submitted manuscript and includes a marked-up manuscript version.

**Referee # 1:**
*The paper is interesting as it offers a report on the use of gliders for performing acoustic surveys to detect and study marine mammals. The specific case present an interesting option based on a low cost recorder rather than custom complex dedicated electronics. However the paper appears more as a basic tech report than a scientific paper.*

**Authors' response:**
The main point of the manuscript was to evaluate the use of a glider fitted with two hydrophones for marine mammal population density estimation studies. Most population density estimation studies have been done with data from fixed sensors, either single sensors or hydrophone arrays. Detection, classification, and sometimes tracking and localization are inherent components of population density estimation from passive acoustics. The intent was to show what extra information or constraints a glider with two phones would provide to such studies and ultimately to adapt the existing density estimation methodology from fixed sensors to moving platforms. We also note that the described experiment was opportunistic and by no means designed as a density estimation experiment. We are making sure those points are stressed and clear in the manuscript. Finally, since Ocean Science is carrying a special issue about the experiment and the use of gliders, we thought that would be the most appropriate venue to submit our manuscript.

**Author's change in manuscript:**
A paragraph was added to the introduction with a summary on marine mammal population density estimation and its extension to data sets collected by gliders, which is a current and on-going research topic. Moreover, the work's objectives were also stated more clearly in that section. Even though the manuscript may seem like a tech report given the description of the experiment and the recording system we used, it also presents novel results derived from the two-sensor data set. This is the first time a glider with two hydrophones has been used to study marine mammals, and the first time animal tracks from estimated bearing angles have been presented. The contributions of this work have also been stated in the discussion section.

**Referee # 1:**
*The findings have no scientific relevance for marine biology and the authors show little expertise in the description of detected biological sounds. Dolphin clicks and sperm whale clicks are well known now. The figures don't show the characteristics of detected events in detail, e.g. to clearly show the differences among artifacts and real signals, or*

*to show the multi paths underlined in the text.*

**Authors' response:**
Since, as the reviewer points outs, dolphin and sperm whale clicks are well known, we did not think it was necessary to present a detailed description of those. Moreover, as stated above, the purpose of the study was not to simply detect and classify marine mammal sounds. However, more or better figures could be easily included to show some characteristics of the recorded data outlined in the text.

**Author's change in manuscript:**
Descriptions and illustrations of all the different sounds observed in the recorded data were added to the "Data Processing and Analysis" section. These include electronic noise and glider self-noise as well as marine mammal sounds. We stress once more that the objective of the manuscript was not to simply detect and classify marine mammals. Given the sampling frequency of the equipment used, and the presence of easily detectable and classifiable sperm whale regular clicks, we chose to focus on those calls for the rest of our analysis. Reported characteristics of sperm whale clicks and their distribution in the Mediterranean were added to the manuscript for completion.

**Referee # 1:**
*The multi paths in recording biosonar clicks is well known and the multi paths can be positively used to improve the localization of sperm whales. Surface multi paths are generated by the sea surface, but often also the sea bottom generates reflections of sperm whale clicks. With a flat sea surface reflected clicks show phase inversion, described in the text as mirror images.*

**Authors' response:**
Multipath occurrence, of any underwater signal will depend on the geographic location, water column structure, and depth of source. In the case of marine mammal calls we don't know where they are, neither in depth nor distance from the recording sensor. Multipath can sometimes be used to aid in localizing whales. However, in order to automatically distinguish multipath in the recorded data, highly specialized algorithms are necessary. Another option is for a human analyst to manually check the data, which can be a time-consuming task. For density estimation studies, detectors of simple characterization are preferred. Therefore, the use of complex algorithms for selecting only direct arrivals was beyond the scope of this work. Our intent was not to localize animals; being able to resolve tracks is sufficient and less time-consuming for density estimation purposes. The term "mirror image" was used to describe the pattern observed in the estimated tracks shown on the bottom plot of Figure 8. We hence assumed they were likely caused by multipath, which upon visual inspection of the corresponding data proved to be true.

**Author's change in manuscript:**
While addressing reviewers' comments, the mirror image pattern described in the manuscript and its association with the occurrence of observed multipath in the data was further investigated. As it turned out, no correlation was found between the two. In fact,

from manual inspection of the automatic detections, it was observed in many instances that the detector considered first (direct) and second (multipath) arrivals as a single detection. Furthermore, by estimating bearing angles of direct arrival and corresponding multipath no difference was observed between the two, i.e., they were coming from the same direction.

In order to avoid misunderstandings, the term "mirror image" was removed from the manuscript. The text and bearing angle figures were updated, and a note was made that multipath clicks had no influence on the results. An example of multipath data, in the form of spectrogram and waveform, was added to the section describing the observed marine mammal sounds since they were a feature observed in the data set.

**Referee # 1:**
*Advantages/disadvantages of the use of a glider are not presented.*

**Authors' response:**
The last paragraph of the introduction lists some advantages and disadvantages of working with gliders for marine mammal studies.

**Author's change in manuscript:**
Small changes were made in the text, specifically to the second to last paragraph of the introduction, to stress the listed advantages and disadvantages of using a glider for marine mammal density estimation studies.

**Referee # 1:**
*Which is the impact of flow noise? How the change in depth influences the recording? Which types of noises are made by the glider itself, e.g. when it changes its asset?*

**Authors' response:**
The sources of noise from a Slocum glider were well characterized by Kristy Moore in her thesis dissertation in 2007. Flow noise was shown to possibly affect frequencies up to 2 kHz, on a 20 kHz sampling frequency system. As we were mostly concerned with higher frequencies, flow noise was deemed not important for our application. Other noise types made by the glider include fin steering, movement of the battery, volume piston, and air pump. These are however, discrete events that do not interfere with the overall acoustic recordings and can be easily distinguished. A note about the flow and other glider noises, including the above-mentioned reference, is being added to the manuscript for completeness.

Glider depth changes would influence the recordings, again depending on the environment (bathymetry and sound speed profile) and the location of the source (whale). Transmission loss and ray calculations are being made with the local bathymetry and sound speed profile recorded by the glider at the same time the acoustic recordings were made. Such information will be added to manuscript to highlight the acoustic environment.

**Author's change in manuscript:**

A subsection on the types of noise produced by the Slocum glider, including examples extracted from the recorded data set, was added to the manuscript as noted on the author's response to referee # 1. A brief section describing the acoustic environment where the data was recorded was also added. The objective was to show, through modeling, how detections could vary with depth.

***Referee # 1:***
*Is the quality of the recorder well suited to the task? Authors write about clicks with energy content increasing with frequency. Most dolphins do produce clicks with peaks above 40 kHz and up to 100 kHz and more. Recording them at close range may result in very high frequency levels that may saturate the hydrophone, its preamplifiers and even the recorder input. Also to consider the resonance of the ceramics in the hydrophones and the possible aliasing effect induced by the intrinsic a-a filters of the recorder that may "reflect" the acoustic energy above Nyquist down to the recorded range.*

**Authors' response:**
We do believe the quality of the recorder was well suited to the task given its high sampling frequency (96 kHz), good bit resolution, and low self-noise. It should be kept in mind that no specific species were initially targeted and that the experiment was opportunistic. While we do understand that 96 kHz sampling frequency may not be enough to capture all frequencies of, for example, dolphin clicks, it is still enough to detect dolphins, potentially classify some of them, and detect other whale species such as sperm whales.

**Author's change in manuscript:**
No specific changes were made in the manuscript regarding this comment.

***Referee # 1:***
A minor point concerns the choice of the recorder. External batteries have been used. Other pocket recorders have less noise and require much less power than the Tascam. Some can run for 48 hours on their two internal AA batteries. The recorder is called "voice recorder" but it should be called "music recorder".

**Authors' response:**
The choice of the recorder was made due to its good specifications and our limited budget. The Tascam offered an inexpensive option with good resolution and high sampling frequency (96 kHz). As shown in Figure 1 (b) of the manuscript only the main board of the original product was used. The plastic cover (which took unnecessary space inside the glider's science bay) was removed, therefore external batteries had to be used to power the device. In its original configuration, the Tascam took two AA batteries and recorded sounds by default at 44.1 kHz at 16-bit resolution. Therefore, in order to record at 96 kHz and 16-bit resolution we found that we needed 8 AA batteries to power the unit in order to record for 24 hours. Due to its construction, the Tascam did not allow recordings past 24 hours. A noise assessment of the Tascam was made when it was first acquired. It showed higher self-noise at lower frequencies (< 1 kHz), but not deemed

sufficiently high to consider it a problem. Research for off-the-shelf recorders at the time (2013-2014) indicated that the Tascam offered the highest sampling frequency, while other pocket recorders had sampling frequencies only up to 44.1~48 kHz.

**Author's change in manuscript:**
The term "voice recorder" was substituted through out the manuscript by the more appropriate "digital recorder." Other original configurations of the Tascam as noted above were added to the text for completeness.

*Referee # 2:*
This manuscript is worthwhile publishing only because the use of acoustics on glider for marine mammal detection is in its infancy and it's important to share various investigators' experiences and results from their field test. In this case, 23 hours of recordings were achieved but most of the data occurred in a 1-hour time span.

**Authors' response:**
After receiving the two anonymous reviews to our manuscript, it became clear to us that the objectives of our work should be more explicitly stated. The main objective was to evaluate the glider data for population density estimation studies, which require all of the components mentioned by both reviewers, such as localization and detection. It was not our intention for the paper to address any singular component, but to present a comprehensive report about all factors. For clarification, we have modified the text to reflect this, and have properly identified all components.

To clarify the data set and the portion we chose to present: because our intention was to demonstrate the type of analyses that could be done with the data and not describe the data in its entirety, we chose the period with the best data. While almost 23 hours of recordings were made by the acoustic acquisition system on the glider, the specific 1.5-hour span was when most of the marine mammal activity was observed. This does not imply that there were no data on the remainder of the recordings. In fact, the absence of detected calls is important in population density estimation.

**Author's change in manuscript:**
As stated above, the work's objectives were made clearer in the introduction. We also made it clearer, on the data analysis section on marine mammal sounds, why we chose the data we showed in the manuscript.

*Referee # 2:*
However, I do have a number of misgivings about this manuscript and are basically involve the avoidance of serious discussion about the usefulness and accuracy of the results. I will details some of the items that the authors should address in a revision.

1. The accuracy of the bearing estimates is never discussed and I think it needs to since I believe the accuracy was not very high. The baseline is too short and the further out the animals are from the glider the more inaccurate the estimate. Also the position of the animals with respect to the glider direction will have a big effect on the accuracy. The

dynamics of the glider, especially the yaw, is not even mentioned. The localization is discussed in a manner that suggest no problems, not issues, perfect localization. I think this issue is considerably more important than the techniques used for localization since time of arrival difference based cross correlation analysis is fairly routine.

**Authors' response:**
The reviewer's comments are well founded and are being addressed in the revised manuscript. With regards to accuracy, a more detailed analysis is being added to the bearing estimation section. The accuracy of the estimate depends on a few things. One channel is used to detect clicks. A small time window around each detection is cross-correlated with the same time window corresponding to the other channel. Cross-correlation gives an estimate of time difference of arrival. So, one can talk about accuracy of the detector, and accuracy of the cross-correlation algorithm. Accuracy can also be thought of as how well one can distinguish two closely vocalizing animals. It also depends on the minimum signal-to-noise ratio between call and background noise levels necessary for the cross-correlation to return a reliable estimate.

The further a vocalizing animal is from the hydrophones, the less likely it will be detected. The environment also plays a big role in how a sound will travel from source to the receivers. Navigation, spatial location and environmental data collected by the glider, as well as propagation modeling results are also being added to the manuscript to provide more insight into detections made and the tracking results obtained.

**Author's change in manuscript:**
Section 5 in the manuscript, which describes the marine mammal bearing tracks, is now divided into three subsections: Bearing estimation, bearing results, and bearing accuracy. The issues regarding accuracy are discussed and an analysis in terms of click SNR is presented. Furthermore, estimated bearings have been corrected taking into account the heading of the glider, and are now plotted also as a function of the peak of the cross-correlation. A low cross-correlation peak indicates low SNR of detected clicks and therefore, a higher error in angle estimation. Despite the errors associated with cross-correlation, the results still suggest the presence of a few tracks.

*Referee # 2:*
2. There is some hand waving in the statement "Such information can be valuable to density estimation methods, either directly for estimating the percentage of time a species produces sound during one day (Marques et al., 2013)." If you have a moving platform and come across a group of animals also moving, directly estimating the percentage of time a species produce sounds can surely be done but what does it mean? How such (bearing estimate) information be valuable to density estimation methods seems like a good statement to make but is it really true with poor bearing accuracy?

**Authors' response:**
The more information available, the better the density estimates since there are more covariates added to the analysis. A more detailed explanation of how the data from two sensors can be used to improve density estimates is given with references. It should be

noted that some aspects of the methodology, like deriving the detection function for a glider, is a current research topic, which we also wish to address in the future.

**Author's change in manuscript:**
A summary on population density estimation is now given in the introduction. Most of the methodologies that have been developed were based on data sets from fixed sensors. Research efforts are currently being made to extend those methodologies to gliders, which are not only moving but are also slower than the marine mammals themselves. However, a study on terrestrial population density using acoustics has shown increased accuracy when using bearings to calling animals.

*Referee # 2:*
3. There should be a better way of displaying click signals then a spectrogram. All you see is a line going to very high frequency (off the chart in some cases) and that's support to tell me more than the time of occurrence? How's about plotting center frequency or peak frequency instead?

**Authors' response:**
Spectrograms continue to be the preferred tool used by many marine bio-acousticians, to show snippets of data or detections of marine animal sounds. Spectra are also shown, though in the case of the sperm whales studied in this manuscript, the lower part of the frequency range is sufficiently distinctive, that detailed spectra are not needed to identify the species. Spectrograms give not only the time of occurrence but also the frequency content of the call (vertical axis) as well as its energy content (color bar, usually in dB). If a sound's bandwidth (or frequency range) is bigger than half the sampling frequency of the instrument then they will appear clipped in the spectrogram (or off the chart). In our case, we cannot detect sounds that are above our Nyquist frequency of 48 kHz. However, figures are being added to the manuscript to better present the types of sounds we detected in our data, both biological and electronic.

**Author's change in manuscript:**
More and different spectrograms were added to the manuscript displaying the different types of sounds observed in the data set.

*Referee # 2:*
4. A minor issue is the phrase in the last line of page 3, ": : :.where high frequencies are highly attenuated." I don't know what highly attenuated means? At 30 kHz the absorption coefficient is about 3.9 dB/km and at 15 kHz its about 1.0 dB/km. I don't consider the 2.9 dB/km difference very large in the broader scheme of ocean propagation.

**Authors' response:**
According to the frequency dependent attenuation formula given by Jensen *et al.* (page 35, equation 1.34 on the first edition), at 30 kHz the attenuation is 8.3032 dB/km and at 15 kHz the attenuation is 2.4693 dB/km. The difference is 5.8 dB/km. In terms of a

broadband sound this difference could mean that only the lower frequency components are detected.

**Author's change in manuscript:**
No changes were made to the manuscript regarding this comment.

*Referee # 1:*
5. I don't understand why click ID software such as M3R is not used to try to ID some of the deep diving odontocetes like beaked whales, Risso's dolphins and pilot whales.

**Authors' response:**
The three species mentioned by the reviewer, beaked whales, pilot whales and Risso's dolphins have click center frequencies reported to be over 30 kHz and bandwidths over 30 kHz. Our recording system offered a sampling frequency of 96 kHz, which is not enough to record the whole spectrum of those species' clicks. On the other hand, Sperm whales have clicks with lower frequency content, allowing us to record most of the clicks' energy.

**Author's change in manuscript:**
No changes were made to the manuscript regarding this comment.

[revised manuscript text omitted]

---

## Author Response (AR2)

Dear Topic Editor,

We are grateful for you last suggestions to our manuscript and have addressed them as follows.

The term "Sardinian Sea" was removed from the text and the first instance "Sardinia" was mentioned we also included "Italy" for context and clarification.

The confusion regarding glider CTD measurements and the calculation of sound speed profiles from those measurements was clarified throughout the manuscript. Moreover, the equation used to calculate sound speed was properly noted and referenced.

Finally, the colormap used in all spectrogram figures (Figures 3, 4, 5, 7, 9, 10, 11) as well as the ranges shown in the colorbar were modified so that the noted features of each figure would appear clearer.

A marked-up manuscript is being uploaded as a supplementary material.

We believe that these final changes have improved the overall understanding of the manuscript. Please do let us know if you have any other concerns.

Best regards,

Elizabeth Küsel

[revised manuscript text omitted]